# How Enzymes, Proteins, and Antibodies Recognize Extended DNAs; General Regularities

**DOI:** 10.3390/ijms22031369

**Published:** 2021-01-29

**Authors:** Georgy A. Nevinsky

**Affiliations:** Institute of Chemical Biology and Fundamental Medicine, Siberian Division of Russian Academy of Sciences, 63009 Novosibirsk, Russia; nevinsky@niboch.nsc.ru

**Keywords:** different enzymes and proteins, anti-DNA antibodies, general regularities of DNA recognition

## Abstract

X-ray analysis cannot provide quantitative estimates of the relative contribution of non-specific, specific, strong, and weak contacts of extended DNA molecules to their total affinity for enzymes and proteins. The interaction of different enzymes and proteins with long DNA and RNA at the quantitative molecular level can be successfully analyzed using the method of the stepwise increase in ligand complexity (SILC). The present review summarizes the data on stepwise increase in ligand complexity (SILC) analysis of nucleic acid recognition by various enzymes—replication, restriction, integration, topoisomerization, six different repair enzymes (uracil DNA glycosylase, Fpg protein from *Escherichia coli*, human 8-oxoguanine-DNA glycosylase, human apurinic/apyrimidinic endonuclease, RecA protein, and DNA-ligase), and five DNA-recognizing proteins (RNA helicase, human lactoferrin, alfa-lactalbumin, human blood albumin, and IgGs against DNA). The relative contributions of structural elements of DNA fragments “covered” by globules of enzymes and proteins to the total affinity of DNA have been evaluated. Thermodynamic and catalytic factors providing discrimination of unspecific and specific DNAs by these enzymes on the stages of primary complex formation following changes in enzymes and DNAs or RNAs conformations and direct processing of the catalysis of the reactions were found. General regularities of recognition of nucleic acid by DNA-dependent enzymes, proteins, and antibodies were established.

## 1. Introduction 

DNA- and RNA-dependent enzymes, proteins, and antibodies play a vital role in many key cellular processes—transcription, replication, recombination, repair, integration, chromosome dynamics, and protection from viruses and bacteria. Therefore, understanding the molecular mechanisms of DNA-dependent enzymes, proteins, and antibody action is critical from fundamental and applied points of view.

It is evident that X-ray analysis data on the structures of complexes between enzymes (proteins) and DNA (or RNA), including conformational changes during their interactions, play an important role in the understanding of mechanisms of nucleic acid recognition [1,2,3,4,5,6,7,8,9,10,11,12,13,14,15,16]. However, X-ray analysis data cannot provide quantitative estimations of different molecular contacts relative importance, or the relative contributions of weak, moderate, or strong specific and unspecific contacts to the total affinity of proteins for nucleic acids (NAs). X-ray analysis of sequence-specific enzymes with DNAs or RNAs sometimes leads to a misunderstanding of the real role of some specific contacts. There is a point of view that the specific contacts between proteins and DNAs that are revealed by X-ray analysis can provide a high affinity for specific nucleotides or sequences of nucleic acids (reviewed in [11,12,13,14,15,16]). There are only a few literature data on the quantitative estimation of the relative individual contributions of individual-specific nucleotides and sequences to thermodynamic (formation of complexes) and kinetic (*k*_cat_, constant rates of reaction) steps to the affinity of enzymes and proteins for DNAs or to the specificity of enzymes actions [11,12,13,14,15,16]. However, only a detailed quantitative estimation of the relative contributions of all nucleotide units of NAs can provide a correct interpretation of data obtained using an X-ray structure analysis. Therefore, it has been have developed a special approach—stepwise increase in ligand complexity (SILC)—to estimate the relative contributions of all individual nucleotides or specific sequences and their various structural elements to an enzyme’s affinity for long DNAs [11,12,13,14,15,16]. The peculiarity of this approach is the gradual complication of the structure of DNA ligands, starting with the possible minimum ligand (structural elements of DNA), which can form any bonds with proteins or enzymes according to the scheme: 



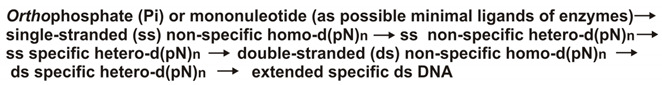



Using the SILC approach, we analyzed replication, restriction, integration, topoisomerization, six different repair enzymes (uracil DNA glycosylase, Fpg protein from *E. coli*, human 8-oxoguanine-DNA glycosylase, human apurinic/apyrimidinic endonuclease, RecA protein, and DNA ligase), and NA-recognizing proteins (RNA helicase, lactoferrin, lactalbumin, human serum albumin (HSA), and IgGs against DNA).

Analysis of various interactions of these enzymes, proteins, and antibodies with long nucleic acids by the SILC approach has shown that the formation of contacts between them and specific nucleotides or sequences of NAs cannot provide their observed high affinity for DNAs and RNAs. Actually, all nucleotide units “covered” by the DNA-binding clefts interact with enzymes and proteins. High affinity is mainly (5–8 orders of magnitude) provided by many weak, unspecific, additive interactions between the enzymes and proteins with different structural elements of many nucleotide units of DNAs and RNAs, mostly with internucleoside phosphate groups and only sometimes with hydrophobic bases [11,12,13,14,15,16]. The relative contribution of specific contacts to the total affinity of DNAs is rather small, not exceeding one, and rarely reaches two orders of magnitude. The enzyme specificity is provided by the enzyme-dependent stage of adjustment of DNAs or RNAs to their optimal conformations, which provide an increase in *k*_cat_ values by 4–8 orders of magnitude ongoing from non-specific to specific DNAs.

## 2. DNA Polymerases

The first enzymes to study general mechanisms of DNA recognition were DNA-dependent DNA polymerases of pro- and eukaryotes [17,18,19,20,21]. These enzymes recognize the template-primer complex.

The Gibbs free energy (∆G°) for the complexes of ligands with enzymes and proteins are usually equal to the sum of the ∆G° values for all individual contacts formed
∆G°_sum_ (corresponding to all *n* contacts) = ∆G°_1_ + ∆G°_2_ +… + ∆G°_n_(1)
where all ∆G°_1-n_ corresponds to all individual contacts of the ligand and ∆G° corresponds to each individual contact = −RT × ln*K*_d_ [22]. The total *K*_d_ value characterizing the formation of any protein–enzyme complex with the ligand is the product of *K*_d_ values for all individual contacts
*K*_d_ (total) = *K*_d_(1) × *K*_d_(2) × *K*_d_(n)(2)

The mechanism of DNA recognition by DNA polymerases (and other enzymes and proteins) was analyzed at the molecular level using the SILC approach in accordance with the following scheme: *ortho*phosphate (Pi) or mononucleotide (as possible minimum ligands of enzymes) → single-stranded (ss) non-specific homo-d(pN)n → ss non-specific hetero-d(pN)_n_ → ss specific hetero-d(pN)_n_ → double-stranded (ds) non-specific homo d(pN)_n_ → ds specific hetero d(pN)_n_ → extended specific ds DNA.

For estimation of different ligands affinities to the template binding sites of DNA polymerase from *E. coli* (Klenow fragment) and human placenta DNA polymerase alpha was used Pt^2+^-containing oligodeoxyribonucleotide (Pt^2^ -ON) [17,18,19,20,21]. The *K*_d_ values for all ligands were estimated from inhibition of polymerase activities inactivation with Pt^2^ –ON affinity reagent interacting with template binding sites with high affinity. All *K*_d_ values obtained are provided in Appendix A. These data were presented in the form of logarithmic (-Log) dependencies of the *K*_d_ values on the number of d(pN)_n_ mononucleotide units (*n*). As an example, Figure 1A shows the data for the DNA polymerase I from *E. coli*. Such dependences were linear at *n* = 1–20 for both DNA polymerases, which indicated the additivity of free energies characterizing enzyme interactions with individual nucleotide units of d(pN)_n_ [17,18,19,20,21]. The *n* values corresponding to the affinity changes and reaching a plateau at *n* = 20 correlated with the relative size of enzyme globules and the length of DNA fragments protected by the enzymes from their hydrolysis by DNases [17,18,19,20,21].

It was finally shown that the interaction of single-stranded (ss) DNAs with template binding sites of DNA polymerases leads to an increase in the DNA affinity due to interaction with one nucleotide units with different bases of d(pN)_n_ by the factor *F*, which include C (1.58), T (1.78), G (1.95), and A (2.0) [17,18,19,20,21]. The *K*_d_ value is determined by a superposition of weak additive non-specific electrostatic and hydrogen bonds, and hydrophobic and/or Van der Waals interactions with individual nucleotides is described by a decreasing geometric progression [17,18,19,20,21]
*K*_d_[d(pN)*_n_*] = *K*_d_[(P_i_)] × (*E*)^1−*n*^ × (*h*_C_)^−*c*^ × (*h*_T_)^−*t*^ × (*h*_G_)^−*g*^ × (*h*_A_)^−*a*^(3)
where *K*_d_ [(Pi)] − *K*_d_ stands for the minimal *ortho*phosphate ligand. The electrostatic factor *E* = 1.52 reflects the increase in affinity due to an interaction of one internucleoside phosphate group of DNA with DNA polymerase template binding site [17,18,19,20,21]. Factor *F* is the product of factors *E* and *h*_N_ (*F = E* × *h*_N_). The *h*_C_ = 1.04, *h*_T_ = 1.17, *h*_G_ = 1.28, and *h*_A_ = 1.32 values correspond to hydrophobic factors *h*_N_, reflecting the increase in the efficiency of the enzyme interaction with one of the bases (C, T, G, or A) of d(pN)_n_, the number of which in this ligand is *c*, *t*, *g*, and *a*, respectively [18,19,20,21]. For both homo- and hetero-oligonucleotides, the contributions of internucleoside phosphates (and different bases) to the affinity of ss DNAs to the template binding sites are the same. The theoretical *K*_d_ values calculated using Equation (3) and the *K*_d_ values obtained from experiments are nearly completely coincided [18,19,20,21,22]. When passing from one to another DNA polymerase, only very slight changes in the numerical values of *K*_d_ [(Pi)], factors *E,* and *h*_N_ are observed [18,19,20,21,22].

It was shown that only one DNA internucleoside phosphate group has a high affinity for DNA polymerases (*K*_d_ = 26 µM, Klenow fragment) due to forming one Me^2+^-dependent electrostatic contact (∆G° ≈ −1.2. kcal/mol) and one hydrogen bond (∆G° ≈ −4.7 kcal/mol) with the enzyme. At first glance, all electrostatic contacts 19 of 20 internucleoside phosphate groups and 20 hydrophobic interactions of bases with template binding sites of DNA polymerases are very weak. However, these weak contacts with the enzyme template-binding site, due to their additivity, provide the whole ∆G° ≈ −4.6 kcal/mol [18,19,20,21,22].

Complementary to templates, primers play a special role in the primer–template complex interaction with DNA polymerases. The *K*_m_ values for d(pN)_n_ primers were measured as functions of their lengths (Appendix A) [21,23]. It was found that all mononucleotides (dNMPs) can serve as minimal primers of DNA synthesis. The −Log dependencies of *K*_m_ values increased linearly until *n* = 10 nucleotide units (Figure 1B) [21,23]. The increase in the length of d(pT)_n_ and d(pA)_n_ primers by one nucleotide link led to an increase in their affinity by 1.74-fold, while d(pC)_n_ and d(pG)_n_ by a factor of 2.33. Considering two hydrogen bonds between A and T bases of the template and primer and vice versa, or three such bonds between C and G from the 1.74 and 2.33 values square and cube roots respectively were extracted, giving 1.32 value. It was shown that only one 3′-terminal nucleotide of the primer forms strong contacts with DNA polymerases, while 9 of its remaining units only with the complimentary template. The factor 1.32 reflects the increase in the primer’s affinity for DNA due to the formation of one Watson–Crick hydrogen bond between the template and the primer. The contribution to the affinity of some internucleoside phosphate groups of the template and primer was estimated using oligonucleotides ethylated at phosphate groups [17,18,19,20,21]. Finally, using all the data obtained, including those presented in Figure 1, the thermodynamic model, i.e., the template–primer binding with DNA polymerases, was proposed and compared with available crystallographic data (Figure 2) [21,23].

Later, the mechanisms of DNA recognition by various enzymes and proteins were investigated. It turned out that all enzymes possess one site of an increased affinity for various dNMPs and for only one nucleotide unit of lengthy DNA and they are capable of recognizing both specific and unspecific DNA with high affinity. Equation (3) has been shown to be applicable to describe the interaction of any ss DNAs with any sequence-independent, and non-specific DNAs with any of sequence-dependent proteins and enzymes. When passing from one to another protein or enzyme and from ss to ds DNA, only a strong change in the numerical values of *K*_d_ [(Pi)] and a slight change in the factors *E* and *h*_N_ are usually observed [11,12,13,14,15,16].

## 3. Uracil DNA Glycosylase

Uracil DNA glycosylases (UDGs) form a ubiquitous, highly conserved specific class of DNA repair enzymes of prokaryotic, eukaryotic organisms, in addition to poxviruses and herpesviruses. They specifically remove uracil base from modified DNA [24,25]. UDGs hydrolyze the glycosidic bond between the deoxyribose sugar and the U base. The resulting abasic site is removed then by apurinic/apyrimidinic endonuclease (APE) and a deoxyribophosphate lyase. The formed gap is filled up by the action of DNA polymerases and DNA ligases. UDGs demonstrate high specificity for efficient removal of uracil from ss and ds DNA, with faster excision from ss DNA [25,26,27].

The general patterns of non-specific ss DNA recognition by UDG were the same as for DNA polymerases. The −Log*K_d_* dependencies on the number of nucleotide units (*n*) in the case of UDG for ss d(pN)_n_ stop increasing at *n* = 9–10, indicating that UDG globule covers only 9–10 DNA base pairs (Figure 3A) [26,27]. Similar to DNA polymerases, the minimal ligands of UDG were *ortho*phosphate (*K*_d_ = 1.7 × 10^−2^ M) and various dNMPs having different affinity to the enzyme (3.3–15 × 10^−3^ M) (Appendix A).

All data on the analysis of UDG and other enzymes using the SILC approach were compared with the published data on the X-ray analysis of these enzymes [28,29,30,31,32,33,34,35,36,37,38,39,40,41,42,43,44,45,46,47,48,49,50,51,52]. The main regularities of extended DNA recognition by DNA polymerases and uracil DNA glycosylase were compared with those in the case a large number of other enzymes and proteins [53,54,55,56,57,58,59,60,61,62,63,64,65,66,67,68,69].

UDG formed weak additive contacts with both internucleoside phosphate groups (*E* = 1.35) and bases *h*_C_ = 1.08, *h*_T_ = 1.22, *h*_G_ = 1.35, and *h*_A_ = 1.42. The affinity of UDG for ss d(pA)_n_ is ~1.5-fold higher than for ds d(pA)_n_ × d(pT)_n_ (Figure 3A) [25,26,27]. This is because UDG partially melts base pairing in ds d(pN)_10_, and its contacts with two strands of ds DNAs almost independent [5,28]. According to X-ray data, the structure of the sugar–phosphate backbone of DNA after the binding of UDG with DNA is drastically changed [5,28]. A conservative Leu-272 amino acid of the DNA-binding cleft is positioned in UDG’s active center; this Leu intercalating with hydrophobic bases locally melts the DNA helix. A further transformation of DNA structure under the action of the enzyme is related to its melting and destruction of stacking interactions in the double helix covered by the enzyme. Then, the uracil residue “flips outside” [5,28]. As it will be shown below, various enzymes recognize DNA according to similar regularities, but changes in the conformation of DNAs leading to specific catalytically competent states occur in very different ways including deformation of DNA backbone—stretching or compression, partial or complete DNA melting, bending or kinking, eversion of nucleotides from the DNA helix, etc. (see below). 

Among all dNMPs, only dUMPs can exist in two almost thermodynamically equivalent conformations, one of which provides the stacking of the bases in ds DNA and the other does not [13,14]. Thus, only UDG can relatively easily melt ds dU-DNAs in comparison with non-specific DNAs promoting effective flipping of the uracil outside the DNA. Only after the uracil residue enters a pocket for its binding, the catalytic process becomes possible.

Considering all weak and strong unspecific and specific contacts, the interaction of UDG with unspecific DNAs was described using a thermodynamic model (Figure 4A). According to X-ray analysis [5,28], the DNA dU unit forms five hydrogen bonds with UDG (Figure 4B). However, all these hydrogen bonds between UDG and specific U-base of DNA provide only ~1 order of affinity (∆G° ≈ −1.4 to −1.8 kcal/mol, (Figure 4B) [26,27]. According to the thermodynamic analysis of modified uracil’s contribution using analogs of an oligonucleotide containing 2′-fluoro-2′-deoxyuridine, its interaction with UDG was characterized by ΔG° < −2 kcal/mol [29].

UDG specifically recognizes the uracil base (∆G° ≈ −1.4… −1.8 kcal/mol; Figure 4B), but the contribution of its weak additive unspecific interactions with ss DNA links is quite high (*K*_d_ = 1.0 × 10^−5^ M; ∆G° ≈ −5.9 kcal/mol; Figure 4A). UDG’s affinity for specific ds or ss homo- and hetero-oligonucleotides (ODNs) depending on their sequences, is only ~10–30 times higher than for non-specific ODNs [26,27]. The efficiency of excision of A-, C-, T-, and G-bases by UDG is about five orders of magnitude lower than U-base [26,27]. Thus, the stage of complexation of UDG with specific DNA provides only approximately one order of specificity of its action. The formation of specific contacts between different enzymes and cognate DNAs, including UDG, is not very important on the complexation stage but exclusively important on the stage of DNAs and enzyme conformations adaptations [11,12,13,14,15,16,26,27]. This adaptation stage provides very precise alignment of the electronic orbitals—“orbital steering” of the reacting atoms and it occurs only in the case of specific DNAs. The specificity of UDG action is provided by the stages of the UDG-dependent dU-DNA adaptation to the optimal conformation, which leads to the increase in catalytic constant (*k*_cat_) by ~5 orders of magnitude in comparison with unspecific DNA [26,27]. It is interesting that the stage of complexation of a large number of enzymes provides only 1–2 orders of magnitude of their specificity, while in the case of specific DNA the *k*_cat_ increases by 4–7 orders of magnitude (see below) [11,12,13,14,15,16].

## 4. 8-Oxoguanine DNA Glycosylases

Oxidative damage to the cell including different pre-mutagenic modifications of DNA bases is important for both carcinogenesis and aging [30,31,32,33,34]. One of the most common DNA lesions is 7,8-dihydro-8-oxoguanine (8-oxoG). During the synthesis of DNA, the appearance of 8-oxoG leads to a replacement of a G:C pair with a T:A pair [35,36,37,38]. 8-oxoG is removed from DNA in *E.coli* by Fpg protein [39,40,41,42,43,44]. In addition to its *N*-glycosylase activity, Fpg has a nicking activity, cleaving both the 3′- and 5′-phosphodiester bonds at an apurinic/apyrimidinic (AP) site by successive δ- and β-elimination, resulting in both the 3′- and 5′-end of the gap phosphorylated [45,46].

The lesion 8-oxoG is removed from DNA in human cells by oxoguanine-DNA glycosylase (hOGG1). Fpg protein is not homologous to hOGG1 [47,48], and the X-ray structure of hOGG1 and its complexes with DNA also have no similarity with Fpg [50,51,52,53]. Human OGG1posses only two catalytic activities—DNA glycosylase and a lyase eliminating the 3′-phosphate at the resulting abasic site [53]. Thus, it was interesting to compare the thermodynamic aspects of 8-oxoG recognition by Fpg and hOGG1.

SILC analysis of oxoguanine-DNA glycosylases was described in [54,55,56,57,58,59,60,61,62]. The minimal ligands of Fpg [55,56,57,58,59] (Appendix A) and hOGG1 [60,61,62] are *ortho*phosphate (1.0 × 10^−2^ and 6.8 × 10^−2^ M), respectively) and various dNMPs ((3.3–8.3) × 10^−3^ and (1.5–9.4) × 10^−3^ M, respectively), having comparable affinity to these enzymes (Appendix A). In contrast to DNA polymerases (Figure 1A), the affinity of unspecific ss homo- and hetero-ODNs for Fpg [55,56,57,58] (Appendix A) and hOGG1 [59] did not remarkably depend on the oligonucleotide bases. The −Log*K*_d_ dependencies for d(pR)_n_ oligomers containing only the sugar–phosphate backbone (R is a tetrahydrofuran analog of abasic deoxyribose) were comparable with those for all d(pN)*_n_* containing different bases (Figure 3B,C). In the case of Fpg, the factor *E* is equal to 1.56 [55,56,57,58,59]. The affinity of specific 8-oxoG units within specific ss and ds DNA for Fpg protein is comparable with that for the enzyme’s affinity for free 8-oxo-dGMP [58]. The affinity of specific ss and ds ODNs for Fpg was ~10-fold higher than for unspecific ones (Appendix A).

The *K*_d_ values of hOGG1 for different ligands are presented in Appendix A. The cognate ss ODN is bound by hOGG1 250-fold stronger than unspecific ss oligonucleotides [59]. In contrast, the affinity of hOGG1 for ds ODN containing 8-oxoG (*K*_d_ = 0.011 µM) is higher than for different ds ODNs containing G by a factor of 790–890, while the difference in the affinity of free dGMP and oxo-dGMP is only 4.1-fold (Appendix A) [59]. Thus, the relative contribution of specific interactions of the hOGG1 with 8-oxoG unit within specific DNA is significantly higher than with free 8-oxo-dGMP. However, one cannot exclude that in contract to Fpg, the specific interaction between the 8-oxoG unit of ss and ds DNAs and the active site of hOGG1 can provide a cooperative effect on the specific DNAs binding. For example, some previously formed unspecific weak contacts with nucleotide units of non-cognate ODNs may probably be greatly strengthened when cognate DNA is bound.

Figure 5 demonstrates the thermodynamic model of Fpg interaction with unspecific DNAs [57], while Figure 6 shows the same for hOGG1. Figure 7 shows the thermodynamic model of hOGG1 interaction with specific DNA [59].

The X-ray structure of non-cognate and cognate hOGG1 × DNA complexes and stopped-flow data indicate that this enzyme distorts any specific and unspecific DNAs, creating a sharp kink (bends into an arc; Figure 5, Figure 6 and Figure 7), nevertheless, it fails to insert the unspecific bases into the active site pocket [49,50,60,61].

As in the case of UDG, the contribution of a cognate 8-oxoG nucleotide to the Fpg affinity for DNA is close to one order of magnitude [54,55,56,57,58]. The affinity of hOGG1 for specific DNA is higher than that for non-specific DNAs by about two orders of magnitude [59]. However, in both cases, the complexation stage cannot provide 8–9 orders of these enzymes’ action specificities. At the same time, the transition from unspecific to specific DNAs leads to an increase in the *k*_cat_ values by 6–8 orders of magnitude. Thus, the specificity of these enzymes’ actions (8–9 orders of magnitude) is realized due to two factors, including the stage of specific DNA recognition (1–2 orders of magnitude) and direct catalysis (6–8 orders of magnitude) [54,55,56,57,58,59,60,61].

## 5. Apurinic/Apyrimidinic Endonuclease

Apurinic/apyrimidinic or abasic (AP) sites can be formed in DNA due to spontaneous base loss, as a result of DNA treatment with different chemicals or physical mutagens (UV or ionizing radiation) and removing damaged bases by several DNA glycosylases [24]. AP-endonucleases recognize AP sites, and cleaving the DNA phosphodiester backbone creates a free 3’-OH terminus suitable for the action of DNA polymerases. The major human apurinic/apyrimidinic endonuclease (APE1) is homologous to the *E. coli* AP endonuclease Xth [62,63,64,65,66]. Crystal structures of wild-type and mutant APE1 bound to DNA containing AP site provide many details concerning possible mechanisms of AP site recognition and its removal due to the backbone significant compression-squeeze for placing the sugar residue of AP–DNA in the active center of the enzyme [62,63,64,67,68]. This is another way of DNA adaptation to a catalytically active state.

According to SILC analysis data, APE1 interacts with D-ribose (*K*_d_ > 0.17 M), *ortho*phosphate (*K*_d_ = 360 µM), different dNMPs (*K*_d_ = 163 µM) as minimal ligands, and 9–10 nucleotide units of ss d(pN)_n_ or base pairs of ds ODNs of different lengths and sequences (Appendix A; Figure 3D). The thermodynamic model of DNA recognition by APE1 is presented in Figure 8.

The affinity of unspecific ss homo- and hetero-d(pN)_n_ for human APE1 depends very moderate on the sequences of ODNs (Figure 3D) [69]. The interactions occur mainly due to weak additive contacts with internucleoside phosphate groups (factor *E* = 1.51) [69]. These non-specific interactions of APE1 with every nucleotide unit provide a high affinity for ss (*K*_d_ = 1.7 µM) and increased by a factor of 5.2 for ds DNA (*K*_d_ = 0.33 µM). The affinity of AP-DNA containing the abasic site is higher compared to unspecific DNA only by one order of magnitude (∆G° ≈−1.1 kcal/mol to −1.5 kcal/mol). Therefore, the formation of the enzyme-DNA complex cannot alone provide the observed specificity of enzyme catalysis. The specificity of APE1, as in the case of other enzymes, lies in the *k*_cat_ value, which is elevated by 6–7 orders of magnitude upon transition from unspecific to specific DNAs [69]. The thermodynamic model of DNA recognition by APE1 is presented in Figure 8.

## 6. DNA Ligase

Mammalian cells contain DNA ligase I and II [2]. We studied using the SILC approach human DNA ligase I [12,13]. Figure 9 demonstrates the *K*_d_ values’ −Log-dependency on the length (*n*) of ODNs for DNA ligase. Like for other enzymes, *ortho*phosphate (*K*_d_ = 1 × 10^−3^ M) and dNMPs are the minimal ligands of this enzyme. The additive interactions of DNA ligase with internucleoside phosphate groups of ss DNAs remarkably stronger (factor *E* = 2.14). The increase in affinity for ODNs and *F* factors up to *n* = 10 proceeds in the same order as for DNA-polymerases with the increase in the relative hydrophobicity of the bases C < T < G < A.

The change in the *K*_d_ values for different d(pN)_n_ is described by a progression
*K*_d_[d(pN)_n_] = [*K*_d_(Pi) = 1 × 10^−3^] × [*e* = 2.14]^−n^ × [h_C_ = 1.1]^−c^ × [h_T_ = 1.29]^−t^ × [h_G_ = 1.51)]^−g^ × [h_A_ = 1.62]^−a^.

The transition from ss to ds DNA leads to an increase of the ligand affinity approximately by one order of magnitude. DNA ligase I recognizes DNA similar to other analyzed enzymes.

## 7. RecA Protein

Homologous recombination is a very important process required to maintain genetic diversity and repair damaged DNA in all living organisms. RecA is the protein of SOS-reparation (a form of induced repair, manifested in the ability of a cell to respond to large DNA damage) playing a central role in homologous recombination in *E. coli*. RecA protein provides a search for homology between two DNA molecules and realizes the exchange of homologous strands [70]. RecA is an ATP-dependent, DNA-binding protein (37.8 kDa) [70]. The process of RecA binding to DNAs occurs in three stages, which are (1) *presynapsis*, during which RecA is polymerized on ss DNA, forming a right-helical nucleoprotein filament; (2) *synapsis*, when ds DNA binds to the *presynaptic* complex, and a search for homology with ss DNA occurs and, finally, (3) *strand exchange*, during which a new DNA duplex is formed and ss DNA is displaced. The binding of RecA protein to ss DNA occurs in a non-specific manner; however, there is a preference for protein filamentation on poly(dT) and on GT-rich sequences [71,72,73,74]. In the presence of ATP, RecA forms a right-handed 100 Å diameter filament with DNA [75]. This filament assembly occurs cooperatively in the direction of 5′→ 3’ ss DNA [76,77]. In a complex with a protein, DNA is stretched by about 50% (another way of DNA adaptation) the distance between nucleotides increases to 5.1 Å and there are three RecA monomers and nine DNA nucleotide residues per filament helix step [75]. Upon binding to ds DNA, the filament parameters are the same as for ss DNA, and the duplex in the complex turns out to be untwisted compared to the B-form of DNA [78,79]. In the absence of ATP, a more compact inactive RecA filament is formed with a step of 64 Å and a distance between nucleotides of 2.1 Å [80].

After filament formation, RecA can bind ds DNA, which interacts with the protein’s second DNA-recognition site. The second RecA center can also bind ss DNA interacting with a filament even more efficiently than ds DNA. The role of the second site of the RecA protein after the homologous strand exchange reaction is to bind the leaving strand after the formation of a new duplex [81].

After binding of ds DNA by the filament, a search for homology between the corresponding strands takes place and the subsequent exchange of strands. Several models have been proposed for the search for homology and chain exchange [82,83,84,85], which have not received further experimental confirmation, however (see below).

To understand the subtle mechanisms of the interaction of any enzymes with specific and non-specific DNAs, it was very important to have quantitative data on the effectiveness of the interaction of enzyme globules with each of the nucleotide links of extended DNA covered by them. The mechanism of the search for homology between DNA strands with a subsequent exchange of strands by RecA was of particular interest. 

It was shown that the first RecA site, similar to other DNA-recognizing enzymes, interacts with *ortho*phosphate (*K*_d_ = 0.5 M) and different dNMPs ((1.2–2.0) × 10^−2^ M)) as minimal ligands (Appendix A [86,87]). The initial filamentation stage proceeds with nearly the same efficiency on ss d(pN)_n_ of different sequences. The filamentation process on the second site of RecA protein is more specific and leads to the formation of stronger contacts of RecA with d(pT)_n_ and d(pC)_n_ as compared to d(pA)_n_ (Figure 10) [86,87].

In the case of some other DNA-recognizing enzymes, the interaction efficiency increases with an increase in the bases’ relative hydrophobicity [11,12,13,14,15,16]. RecA demonstrates an inverse relationship—more hydrophobic d(pA)_n_ ligands interact with the second site of RecA worse than less hydrophobic d(pT)_n_ and d(pC)_n_ oligonucleotides [86,87]. Based on a comparison of the affinity of RecA for d(pA)_n_ with that for ODNs ethylated at internucleoside phosphate groups, and for d(pR)_n_ oligomers containing no bases, it was concluded that, in an extended filament, RecA does not form effective contacts with A-bases, and the affinity d(pA)_n_ is provided mainly only due to RecA interaction with the sugar-phosphate backbones [86,87].

Because the degree of RecA-dependent ATP hydrolysis induced by different d(pN)_20_ correlated well with the efficiency of protein filamentation on different ss DNA, the effect of different bases on the efficiency of the interaction of RecA with various ss poly(N) was assessed by the efficiency of ATP hydrolysis (Appendix A) [86,87,88]. Despite the fact that guanosine has the same O6 acceptor and HN1 donor groups as inosine, poly(dG), such as poly(dA), weakly interacted with RecA. Deamination of poly(dG) or poly(dA) led to an increase in the efficiency of ATP hydrolysis, and, accordingly, to more efficient filamentation of the modified DNAs [88]. The same increase occurred when moving from poly(dAG) to poly(dIX). Mixed purine–pyrimidine DNA interacted with RecA in different ways. Thus, poly(dAC) and poly(dTG) bind well to RecA; poly(dAT) showed an intermediate efficiency between poly(dA) and poly(dT) ligands, while poly(dCG) hardly stimulated RecA-dependent ATP hydrolysis (Appendix A).

It is known that the A-form of RNA is characterized by the 3’-endo-conformation of ribose [89], which might be the main factor of the absence of effective interactions between RecA and RNA [86,87,88]. Since DNA significantly unfolds and stretches by 50% during the filamentation with RecA, the 2′-endo-conformation of the sugar-phosphate backbone should be more preferable for this process, because in this case, the helix is less compact, and the distance between nucleotides along the helix axis is 3.03–3.37 Ǻ, in contrast to RNA, where it is 2.56–3.29 Ǻ [89]. Thus, the preference for one of the ribose conformations allows RecA to distinguish between DNA and RNA, and thereby hinder the homologous exchange only between DNAs [86,87,88].

After binding ds DNA with RecA − ss DNA filament, a homology search between the corresponding strands occurs, which is followed by the exchange. The mechanism of this process is still unclear. Several possible homology search mechanisms have been described in the literature. It was assumed that ss DNA can replace one of the strands through the duplex’s major or minor groove [70]. It is believed that in the first case, before the exchange reaction, a DNA triplex is formed, which was named the R-form of DNA [82,83]. In the second, alternative case, the homology search may be carried out only due to the formation of standard Watson–Crick hydrogen bonds, which are formed after base eversion [84]. It is assumed that because the bound ss and ds DNA molecules are substantially stretched and untwisted, the bases can easily leave the bodies of this DNA and be checked for homology. However, these models have not received further experimental confirmation [82,83,84,85,86,87,88].

Based on the data on the analysis of the processes of the filamentation of the RecA protein on various ss and ds DNA, we proposed a new model describing the formation of specific weak additive hydrogen bonds of the RecA protein with T-, C-, and G-bases, which cannot be in the case of A-bases [86,87,88]. The assumption of the presence in the vicinity of the acceptor C = O groups of pyrimidines and NH_2_- and C = O groups of the G base, in addition to amino acid residues of the RecA filament (OH, COOH, etc.), which are proton donors, made it possible to construct a model for recognizing the nucleotide sequence DNA in the second site, which is linked to the first site of the RecA filament [87,88]. In the presence of appropriate proton-donor filament amino acid residues for any combination of chain homology (A • A • T, or C • C • G; and G • G • C), first the keto-enol and then the amino-imine shifts can occur, which results in reorganization of hydrogen bonds leading to the formation of a new duplex between two DNA strands linked to the first and second sites of the filament (Figure 11) [87,88]. After the strand exchange reaction, the second site binds the DNA strand that is displaced from the homologous duplex, stabilizing the newly formed duplex. It should be assumed that the interaction of ss DNA with the second DNA-recognition center may not have a strictly specific character, since the correct specific interaction between homologous strands of the newly formed duplex is sufficient for homology recognition, which is consistent with our data. In this case, the nucleoprotein interactions at the second site should provide a strong anchorage of the outgoing strand to prevent the reverse reaction, which is consistent with the data of the filament analysis using the rapid kinetic method [90].

## 8. RNA–DNA Helicase

Unlike protein recognizing “spatial structure elements” of definite sequences, the enzymes responsible for reproducing genetic information such as RNA–DNA helicases are functioning independently of the NA primary, secondary, and tertiary structure peculiarities. The role of RNA and DNA helicases in replication is the denaturation of ds NAs leading to the appearance of their ss regions [91]. RNA rearrangement proceeds via protein-induced formation and subsequent resolution of RNA branch migration structures, whereby the latter step is dependent on ATP hydrolysis. Despite RNA helicase more specific in the reaction of the unwinding of ds RNA substrates, it can interact with ss RNAs and with ss and ds DNA ligands. We have studied p68 RNA helicase using the SILC approach [12,13].

Similar to the case of other enzymes, the minimal ligands of helicase were *ortho*phosphate (*K*_d_ = 0.1 M) and different dNMPs (~6.3 × 10^−2^ M). The dependency of −Log*K*_d_ upon the number of nucleotide units (*n*) is given in Figure 12A. The *K*_d_ values change for different d(pN)_1–10_ can be described by the following progression
*K*_d_[d(pN)_n_] = [*K*_d_(*P*i) = 0.1M] × [*E* = 1.61]^−n^ × [*h*_C_ = 1.0.5]^−c^ × [*h*_T_ = 1.26]^−t^ × [*h*_G_ = 1.48)]^−g^ × [*h*_A_ = 1.59]^−a^.

The transition from ss to ds ODNs leads to an increase in the affinity ~10-fold (Figure 12A).

## 9. *Eco*RI Endonuclease

Among the type II restriction endonucleases, the dimeric form of *Eco*RI (31 kDa) is the best-studied enzyme. Endonuclease *Eco*RI cleaves a bond between G and A bases of the hexanucleotide sequence in a duplex DNA shown below.

Points of cleavage: 5′-G ↓ AATT C-3′3′-C TTAA ↑ G-5′

The enzyme crystal structure was determined [1,99,100]. It was shown that the interaction of *Eco*RI with ODNs containing a specific recognition sequence leads to conformational changes in both components. The structural features of the B-DNA double helix of DNA regions outside the enzyme are retained, but there are changes at a specific region associated with the enzyme. The enzyme also undergoes conformational changes of its quaternary structure. Certain amino acid residues of the protein form a so-called arm, which “twists” around the DNA, fixing it on the enzyme. Two symmetric “channels” of two subunits are located at a distance of 3.5 Å [1,99,100]. This seems to be the main factor responsible for specific DNA melting and bending in the central part of the DNA recognition site. In addition to electrostatic contacts with internucleoside phosphate groups of DNA, each of the *Eco*RI subunits forms hydrogen bonds with the bases of specific DNA. Thus, the guanidinium group Arg-200 forms two bonds with N-7 and O-6 atoms of G nucleotide. Glu-144 and Arg-145 interact, respectively, with N-6 and N-7 atoms of neighboring A nucleotide, forming also two hydrogen bonds. A total of 12 hydrogen bonds are formed between the specific ds DNA and the two subunits of protein, which are believed to provide specific recognition of the substrate [1,99,100]. These data are of a qualitative nature. 

The quantitative data were found using the SILC approach (Appendix A), and the dependencies of −Log*K*_d_ values on the length (*n*) of ss d(pN)_n_ and their duplexes are presented in Figure 12B [92,93].

One of 6–7 subsites of *Eco*RI subunits has a high affinity to P_i_ (*K*_d_ = 31 mM) and deoxyribosophosphate d(pR) (*K*_d_ = 4.6 mM) [92,93,101]. This suggests the formation of additional contacts of one *Eco*RI subsite with the sugar residue of the d(pR). The enzyme has an affinity for thymine (*K*_d_ = 450 mM) and thymidine (*K*_d_ = 65 mM). Additivity of ΔG° values is retained: *K*_d_ value for dTMP can be calculated as the product of *K*_d_ values for *ortho*phosphate and thymidine as well as for d(pR) and thymine (Appendix A). Consequently, one specific subsite of *Eco*RI recognizes dNMP (or one nucleotide unit of DNA) by additive interactions with all of its structural elements—sugar residue, phosphate group, and a base. The affinity of *Eco*RI to *ortho*phosphate is about 24-fold higher than that of non-charged triethylphosphate (*K*_d_ = 740 mM), which would efficiently interact with the enzyme only through the oxygen atom of the P = O group. Thus, the enzyme most probably interacts with both oxygen atoms (O = P-O^−^) of the phosphate group of dNMPs as well as with one internucleoside phosphate group of specific DNA [93]. The slopes of the −Log*K*_d_ dependencies on the number of nucleotide units (*n*) for *Eco*RI are nearly coincide for all homo- and hetero-d(pN)_n_ (Figure 12B). The *K*_d_ value for oligomers without bases d[(pR)_n_], where R is a chemically stable analog of deoxyribose, are comparable with *K*_d_ values for d(pT)_n_ and d(pA)_n_ of the same length (Appendix A) [93,101]. Thus, *Eco*RI interacts only with the sugar-phosphate backbone of unspecific DNAs. 

Taking into account X-ray data [1,99,100], one can conclude that about 5–6 subsites of each subunit of *Eco*RI can form weak electrostatic contacts with internucleoside phosphate groups of d(pN)_n_. *Eco*RI use as substrates specific ds DNAs. The second chain slightly decreases the *Eco*RI affinity to ds ODNs in comparison with ss d(pN)_n_. Therefore, it seems that the interaction of two *Eco*RI subunits with DNA chains may significantly weaken the complementary interactions between two chains. The enzyme affinity for different hetero-d(pN)_n_ completely or partially corresponding to the specific DNA sequence was determined (Appendix A) [92,93]. The *K*_d_ values for various short homo- and hetero-ODNs of the same length (*n* = 3–5) were comparable. The increase in the affinity ~10-fold was observed only for ODNs containing all six bases of the specific sequence. These results well correlate with X-ray data [1], according to which *Eco*RI should form 12 hydrogen bonds with the bases of a specific sequence of ds DNA. *Eco*RI is a very sequence-specific enzyme. As mentioned above, most repair enzymes’ affinity for specific DNAs is usually higher than for unspecific ones only by one order of magnitude.

Interestingly, *Eco*RI affinity for specific 16-mer ds ODN is only 50–100-fold higher than for unspecific ds ODNs of the same length (Appendix A). Weak additive non-specific electrostatic contacts of *Eco*RI with phosphate groups of ODNs provide approximately five orders of magnitude in DNA affinity, whereas the contribution of specific interactions (12 hydrogen bonds) is no more than two orders of magnitude of a total ds ODN’s affinity. Thus, as for other enzymes, the complex formation between *Eco*RI and specific ds DNA cannot provide specificity of its action. At the same time, *Eco*RI-dependent cleavage of d(pT)_15_ × d(pA)_15_ is 6–7 orders of magnitude slower than that for a 16-mer specific substrate [92,93,101]. Consequently, the *Eco*RI-dependent catalysis step is more sensitive to DNA structure than the step of the complex formation.

## 10. HIV-1 Integrase

Replication of HIV-1 DNA depends on integrating ds DNA copy of the virus genome into the human cell nuclear genome (reviewed in [102]). The retroviral integrase (IN) catalyzes the DNA integration; the recognition sequence is located at the ends of DNA long terminal repeats (LTRs). This LTR-specific sequence is critical for site-specific DNA cleavage and integration [103,104]. To insert both ends of the viral DNA into the host cell genome, HIV-1 integrase catalyzes two reactions—(1) 3′-processing reaction, when the two GT nucleotides from the 3′-ends of each linear viral DNA strands are removed, leaving the CA dinucleotide at 5-ends and (2) the strand transfer of DNA joining reaction in which the modified viral DNA ends are inserted into the host DNA [105,106]. 

The IN × DNA complex formation was analyzed by the SILC approach [94,95]. First, it was shown that IN recognizes free dNMPs (*K*_d_ = 5 mM) through interactions with all its structural elements (base (>50 mM), sugar (>50 mM), and phosphate (16 mM)), the phosphate makes the major contribution (Appendix A).

All −Log-dependencies for unspecific ss d(pN)_n_ were biphasic (Figure 12C). The transition from dTMP, dCMP, or dAMP to the corresponding dinucleotides leads to the affinity increase by a factor of 2.5, 50, and 62, respectively. IN demonstrated two parts of the dependencies close to linear at *n* = 0–2 and *n* = 4–21, respectively (Figure 12C). The region between *n* = 2–4 has a transitional character. Monotonic decrease in *K_d_* (*n* > 4; *K*_d_
*=* 1*/F* = 0.79–0.33 M)), values characterized the interaction of IN with each d(pN)_n_ nucleotide unit. The affinity of IN for one of the free dNMPs was ~1000-fold higher than that of other nucleotide units at *n* > 4. The linear −Log*K*_d_ dependencies for ss ODNs provide evidence of ΔG° values close to an additive for the interaction of some of the 21 individual units of d(pN)_21_ with the integrase. For HIV-1 integrase, the affinity for unspecific d(pC)*_n_*, d(pT)*_n_*, and d(pA)*_n_* did not follow the relative hydrophobicity of the bases. The affinities of d(pA)_21_ and d(pC)_21_ with minimal and maximal relative hydrophobicities are similar, while the affinity of d(pT)_21_ is significantly lower (Figure 12C; Appendix A). Interestingly, a faster increase in the affinity for d(pC)_n_ as compared with d(pT)_n_ resulted in a difference of ~2 orders of magnitude for their 21-mer ODNs (Figure 12C). The higher affinity of d(pC)_n_ can be explained by a better adaptation of d(pC)_n_ to a specific conformation in the viral DNA × integrase complex since of a very flexible structure of d(pC)_n_, while d(pT)_n_ usually adopts a rigid non-flexible structure [107]. Therefore, we have compared the affinity of IN for d(pT)_n_ and oligothymidylates containing several d(pC) links in different positions (Appendix A). The affinities such ODNs were comparable with that of d(pC)_n_ but not for d(pT)_n_ of the same length. Thus, the flexible structure of d(pC)_n_ may be important for its productive changes during complex formation [94,95].

The 3′-end processing catalyzed by HIV-1 integrase was assayed with the specific substrate of IN ds GT-ODN_21_ shown below.

Point of cleavage:5′- G-T-G-T-G-G-A-A-A-A-T-C-T-C-T-A-G-C-A↓**G-T** -3′3′- C-A-C-A-C-C-T-T-T-T-A-G-A-G-A-T-C-G-T C-A -5′

An analysis of the relative affinity of individual strands and of the specific duplex itself with and without 3′-terminal CT dinucleotide and different length sequences corresponding to both chains of the duplex was carried out (Figure 12C; Appendix A). It turned out that the contribution of GT terminal nucleotides to the total affinity of IN for ds GT-ODN_21_ before the processing reaction is significantly higher than that of any other nucleotide of ds GT-ODN_21_. However, after removing the 3′-GT-terminal dinucleotide, IN forms new strong contacts with four 3′-AGCA-terminal nucleotides of the first CA-ODN_19_ chain of the shortened duplex. With this in mind, the recognition of a specific ds DNA by integrase was described using two thermodynamic models corresponding to the formation of complexes with ds GT-ODN_21_ before (Figure 13) and after removing the terminal GT nucleotide (Figure 14). The affinities of IN for ss GT-ODN_21_ (10 nM) and ss CA-ODN_19_ (30 nM) were nearly the same and only respectively 7.7- and 2.5-fold lower than those of their duplexes (Appendix A). The complex formation provides only for a ~30-fold difference in the affinity between specific to non-specific DNAs. At the transition from unspecific to specific ODNs, the IN-dependent processing reaction’s reaction rate is increased by more than 5–6 orders of magnitude. Thus, the catalytic step is significantly more sensitive to the specific viral DNA sequence than the IN × DNA complexation step.

## 11. Human DNA Topoisomerase I

Eukaryotic DNA topoisomerase I (Topo) changes the topological state both positively and negatively supercoiled (sc) DNA and plays an important role in key biological processes of the cell: replication, repair, transcription, integration, and recombination [108]. Topo consists of the N-, C-terminal, core domains, and a linker region [109]. The N-terminal domain is highly charged [109]; it is responsible for the localization of the enzyme in the nucleus [110] but is not important for the enzyme activity [109]. Topo is not absolutely sequence-specific for DNA sequences, but the below sequence is most preferred; the reaction rate in the case of this sequence is 3–4 orders of magnitude higher than for several other DNA sequences cleaved by the enzyme [111,112]. According to X-ray analysis, human Topo forms a large number of contacts with DNA internucleoside phosphate groups and specifically interacts with the T-base of the cleaved DNA strand:



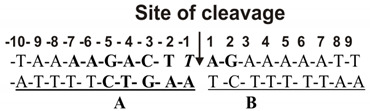



Recognition and transformation of the Topo substrate include the following stages: (a) binding of supercoiled DNA, (b) cleavage of one DNA strand, (c) shedding of supercoils, (d) ligation of the cleaved DNA strand, (e) changing the protein conformation for the following relaxation stages, and (f) dissociation of topo from relaxed DNA [113]. However, the X-ray data qualitative nature does not allow understanding which of these contacts make the main contribution to the total affinity of the enzyme for a specific DNA sequence.

First, we have analyzed the interaction of Topo with unspecific ODNs. *Ortho*phosphate (*K*_d_ = 0.38 M) was a minimal ligand of Topo (Appendix A) [96,97,98]. Figure 12D shows the −Log-dependencies *K*_d_ values on the length (*n*) of the used d(pN)_n_. It was shown that there are 20 DNA base pairs within the protein globule of the enzyme [114]. However, the affinity ss and ds d(pN)_n_ increases linearly up to *n* = 10 (Figure 12D). Data on the Topo’s affinity for different short ONs corresponding to different parts of the specific DNA sequence are summarized in Appendix A. The factor *F* corresponding the increase in the affinity upon d(pN)_n_ lengthening by one nucleotide unit were calculated as 1.71—d(pC)_n_, 1.93—d(pT)_n_, 2.12—d(pG)_n_, 2.33—d(pA)_n_, and 2.89—for duplex d(pT)_n_ × d(pA)_n_, and electrostatic factor *E* is equal to 1.67 (*K*_d_ = 0.6 M, ∆G° = −0.3 kcal/mol) [96,97]. Thus, the *h*_N_ coefficients of increase in Topo’s affinity due to hydrophobic interactions with different bases were calculated (C—1.02, N—1.16, G—1.27, and A—1.40). It is obvious that the relative contribution of additive “electrostatic” interactions exceeds the contribution of “hydrophobic” contacts, and only in the case of d(pA)_n_, they become comparable. Overall, the interaction of Topo with 10 units of ss DNAs is described by the geomantic progression common for all studied enzymes given above.

From the X-ray data [115], the internucleoside phosphate groups of specific ODNs form with Topo from 1 to 4-hydrogen bonds. The value ∆G° = −0.3 kcal/mol is characteristic of weak ion–dipole and dipole–dipole interactions [22]. Therefore, it is reasonable to consider that internucleoside phosphate groups of unspecific ODNs do not form hydrogen bonds with the enzyme and their weak additive interactions with Topo refer to ion-dipole and dipole–dipole interactions. The interactions of negatively charged phosphate groups of unspecific DNAs with a positively charged surface of Topo DNA-recognition site can also be attributed to interactions not of directly contacting groups, but of oppositely charged surfaces of these biopolymers. 

Then, an analysis of the complexation of Topo with other different hetero-oligonucleotides of various structures and lengths corresponding to different parts of the specific DNA sequence was carried out (Appendix A). The transition from unspecific to various specific ODNs, the effectiveness of some contacts increases ~6–30 times, and the *K*_d_ values for several specific nucleotide units decreased to 0.1 to 0.02 M. Figure 15 demonstrates the efficiencies of contacts of several structural elements corresponding to the central part of specific DNA with Topo estimated by us using different ss and ds hetero-ODNs. We came to the conclusion that hydrogen bonds can be formed only after the first step of a specific change in the structure of DNA and Topo. In addition, the efficiency of most hydrogen bonds detected by X-ray analysis [115] may be relatively low. For example, if to assume that all 4 hydrogen bonds between the (-1-2) -phosphate group of AA residues of specific DNA and Topo (Figure 15) are comparable in their efficiency, then the formation of each of them will lead to an increase in the enzyme’s affinity for the ligand only by ~2.1–2.7 times (∆G° = −0.4 … −0.6 kcal/mol). These values are quite comparable with those for weak additive contacts of Topo with internucleoside phosphate groups of ss ODNs. It should be noted that Topo stimulates several successive stages of changes in the conformation of specific DNA to its catalytically active state. For example, the affinity of specific ODNs for Topo increases after their mixtures preincubation for 10–15 min (Appendix A) [116,117].

Based on the analysis of the data obtained for a large number of specific and unspecific, short and long ss and ds ODNs, and the interaction of Topo with supercoiled DNA, the recognition of specific DNA by Topo can be described as follows. 

At the first stage of the formation of the primary complex with unspecific ss and ds DNA, Topo binds these ligands mainly due to weak additive contacts, the affinity of enzymes for these DNAs is quite high; the *K*_d_ values, for ss d(pA)_10_ and ds d(pA)_10_ × d(pT)_10_ are 8.5 × 10^−5^ and 1.0 × 10^−5^ M, respectively [96,97,98].

Thus, the contribution to the affinity of non-specific additive contacts of topo for 10 base pairs of the central part of the duplex can be estimated to be close to five orders of magnitude (*K*_d_ = 10^−5^ M). Strengthening these contacts and/or the formation of additional contacts of the enzyme with 10 base pairs of specific d(pN)_9_ leads to the increase in the affinity by about one order of magnitude (*K*_d_ = 10^−6^ M). Additional interactions of Topo with AT-flanking sequences of extended specific 27-mer ds ODNs increase the affinity by approximately one order of magnitude (*K*_d_ = 10^−7^ M). The increase in the affinity of various unspecific and specific ODNs after their preincubation with Topo provides a change of Topo-DNA complex, resulting in the affinity increase by an additional one order of magnitude (*K*_d_ = 10^−8^ M). Interestingly, the affinity for 27-mer specific ds ODN (*K*_d_ = 10^−8^ M) is comparable to that for relaxed plasmid DNA, while the difference in the affinity between supercoiled and relaxed DNA is approximately two orders of magnitude [116]. Thus, the contribution of specific interactions of Topo with the structural elements of supercoiled DNA (*K*_d_ = 10^−10^ M) does not exceed two orders of magnitude. These interactions with supercoiled DNA may depend on both additional factors of the curvature of the Topo-binding site and on the structural features of supercoiled DNA. In addition, it cannot be ruled out that only supercoiled DNA is capable of forming any additional contacts with nucleotide units distant from the central part of the DNA specific duplex.

In general, the high affinity of Topo for supercoiled DNA is provided by the sum of all the factors listed above, which is 10^−5^ M × 10^−1^× 10^−1^ × 10^−1^ × 10^−2^ = 10^−10^ M (Appendix A) [97,98]. In addition, as in the case of other enzymes described above, the high specificity of the action of eukaryotic topoisomerase I is provided mainly by the catalysis stage. When passing from absolutely non-specific to specific DNA, the reaction rate increases by 5–6 orders of magnitude [96,97,98].

## 12. Human Lactoferrin

Lactoferrin (LF) is a protein with a single polypeptide chain (76–80 kDa) containing two lobes, each of which binds one Fe^3+^ ion and contains one glycan chain [118,119]. Many different functions were attributed to LF—immunomodulation and cell growth regulation, protection from iron-induced lipid peroxidation, transcriptional activation of specific DNA sequences, etc. (for review see [119]). For a long time, LF was considered as a protein that does not possess any enzymatic functions. First, it was shown that LF efficiently splits RNAs [120]. LF is protein interacting with DNA specific sequence and activating transcription [121]. In addition, LF is an enzyme possessing five different catalytic activities— DNase, RNase, ATPase, phosphatase, and amylase [122]. LF possesses two DNA-binding sites that interact with specific and non-specific ODNs [123]. Thus, LF can be considered as a very interesting polyfunctional protein–enzyme possessing several enzymatic functions.

Appendix A demonstrates that the minimal ligands of the first LF DNA-recognizing site are *ortho*phosphate (*K*_d_ = 5 mM), deoxyribosephosphate (*K*_d_ = 3 mM), and different dNMPs (*K*_d_ = 0.56–1.6 mM). The affinity of LF for dCMP is 1.8–2.9-fold higher than for other dNMPs (Appendix A). The affinity for deoxyribose is very low (*K_d_* ≈ 0.6 M). The *K_d_* characterizing of the protein affinity for the C base of dCMP is 0.19 M. Thus, one site of LF recognizes free dNMPs through interactions with all their structural elements (phosphate, sugar, and base); the phosphate making the major contribution to the affinity.

The linear −Log dependencies of *K*_d_ (Appendix A) for ss d(pN)_n_ (0 ≤ *n* ≤ 10–11, *n* = 0 corresponds to *ortho*phosphate), demonstrate the additivity of ΔG° values for the interaction of LF with 10–11 individual nucleotide units of d(pN)_n_ (Figure 16A).

Values of factor *F* and *K*_d_ (2.28 ± 0.02 M; *K*_d_
*=* 0.44 ± 0.004 M) for d(pT)_n_ and d(pA)_n_ are lower than that for d(pC)_n_ (2.36 ± 0.03 M; *K*_d_
*=* 0.42 ± 0.005 M). Thus, the *K*_d_ values reflecting the LF affinity for different dNMPs (0.56–1.6 mM) are approximately 81–770-fold lower than the *K*_d_ (average ~0.43 M) characterizing the enzyme interaction with other from 9 to 10 nucleotide units of an extended ODNs. 

In contrast to DNA polymerases, human UDG, APN1, and DNA topoisomerases but similarly to *E. coli* Fpg, human hOGG1, and *Eco*RI endonuclease, the affinity of LF for different d(pN)*_n_*, does not remarkably depend on the relative hydrophobicity of ODNs bases (Figure 16A). The *E* factors (2.28–2.36) and *K_d_* values (~0.42–0.44 M) characterizing the weak interactions of LF DNA-binding site with each nucleotide unit of different ss d(pN)_n_ were nearly the same. Oligomers with and without bases have nearly the same affinity for LF (Appendix A). The affinity of d(pT)_10_ (0.83 μM) was ~12-fold higher than that for ethylated d[p(Et)T]_10_ (*K_i_* = 10.0 μM) (Figure 16A; Appendix A) that points to an important role of internucleoside phosphates negative charges for LF complexation with DNA. Negatively charged internucleoside phosphate groups of unspecific ODNs could interact with LF as in the case of other various enzymes through dipolar electrostatic forces rather than electrostatic interactions between point charges.

Figure 16A shows that the minimal ODN exhibiting duplex properties toward LF is d(pT)_12_ × d(pA)_12_, for which the melting point is higher than the reaction temperature (25 °C). Thus, there is no effective stabilization of short ODN duplexes at their interaction with LF. The relative affinity of the first DNA chain is ~6–6.5 orders of magnitude, while the addition of the second chain results in an affinity increase of only 16–33-fold. Thus, similar to other enzymes, the contribution of the second DNA strand for LF is significantly less than that of the first one. 

LF interacts with the specific sequence of ds DNAs and activates transcription TAGAAGATCAAA (ODN1; Figure 16A) [121]. The affinity of ss 12-mer ODN1 (*K_d_* = 8.0 × 10^−8^ M) was 4.0–7.9-fold higher than different unspecific ss 12-mer ODNs (*K_d_* = (3.2–6.3) × 10^−7^ M), while ds ODN1 (*K_d_* = 1.0 × 10^−8^ M) demonstrated nine-fold higher affinity than ds homo-d(pN)_12_ of the same length (*K_d_* = 9.0 × 10^−8^ M). Overall, the types of LF interaction with unspecific and specific DNAs can be summarized using the thermodynamic model (Figure 17).

## 13. Human Serum Albumin

Human serum albumin (HSA) is a major protein of human blood possessing very important biochemical and physiological functions [127]. HSA is a very universal protein interacting with many endogenous and exogenous compounds—fatty acids, metabolites, metal ions, pharmaceuticals, and many other blood components. Because of essential diversity in the ligands binding, HSA is characterized by significant structural diversity [127]. HSA has also long been regarded as a protein lacking any enzymatic activities. Later, it was shown that HSA has several catalytic activities (the hydrolysis of amides, esters, and organophosphorus compounds) [127], in addition to RNase and DNase ones [128,129]. However, there were no data on the mechanisms of DNA recognition by HSA. 

It was shown that HSA possesses two DNA-binding sites. The relative contributions of d(pN)_n_ to the total affinity for these two binding sites with lower and higher affinity for DNA were estimated (Appendix A). The minimal ligands of these two binding sites are *ortho*phosphate (*K*_d_ = 3.0 Mm and 20.0 mM in the case of two sites, respectively) [125]. The first binding site discriminates different dNMP, meaning the affinity decrease is in the order (Mole/Liter): dAMP (5.6 × 10^−6^) > dTMP (1.7 × 10^−5^) > dCMP (4.0 × 10^−4^). The second site also discriminates various dNMPs with the order dAMP (6.3 × 10^−5^) > dTMP (1.0 × 10^−3^ M) > dCMP (1.8 × 10^−2^), but it possesses for these mononucleotides significantly lower affinity than the first site, i.e., d(pC) (222-fold) > d(pT) (170-fold) > d(pA) (8.9-fold) (Appendix A). Maximal contribution to the total affinity of all d(pN)_n_ in the case of both sites was observed for their one nucleotide and remarkably lower for three additional nucleotide units of d(pN)_n_ (*n* = 1–4) with a significant decrease in the contribution at *n* = 5–6, and at *n* ≥ 7–8 all dependencies reached plateaus (Figure 16B,C). All d(pN)_n>10_ finally demonstrated high affinity for HSA first (1.4–150 nM) and second (80–2400 nM) sites. The thermodynamic parameters characterizing the contribution of different nucleotide units of all ss d(pN)_1–9_ (ΔG°) to their total affinity for HSA were estimated. Overall, two HSA DNA-binding sites recognize ss DNA by forming additive contacts with 6–8 nucleotide units of ss d(pN)_n_, but in contrast to other enzymes, the relative contributions of d(pN)_n_ nucleotide units (ΔG°) at *n* = 2–8 are not the same (Figure 18) [125].

The second strand’s contribution to the affinity of ds DNA for all analyzed enzymes is usually much smaller than that for the first one. HSA demonstrated a significantly lower affinity for complementary ds d(pN)_n>10_ with T_m_ melting points higher than the temperature used (Figure 16B,C; Appendix A). However, in general, DNA recognition by albumin occurs in accordance with the same regularities as for other enzymes and proteins [125].

## 14. Human α-lactalbumin

Human milk α-lactalbumin (LA, 14.1 kDa) is the major protein of milk [130]. LA does not possess nuclease activities. It consists of a small β-fold and a large α-helical domain. It was showed that α-lactalbumin is a protein that can acquire various functions depending on its folding state [130]. LA interacts with DNA bound with histones in tumor cells and impaired the chromatin structure [131]. In the literature, there was no information on how LA recognizes DNA. 

The SILC approach was used for the analysis of human LA (Appendix A) [126]. The LA DNA-binding site recognizes *ortho*phosphate and all dNMPs (*K*_d_ = (5.0–43) × 10^−5^ M) as minimal ligands [126]. For LA, an increase in the affinity (−Log*K*_d_) was observed only for 5–6 nucleotides of ODNs (Figure 16D). Maximal contribution to LA’s total affinity was observed for three-nucleotide units of all d(pN)_n_ with a considerable decrease in the order 1 > 2 > 3 nucleotides. At *n* = 4–6, the affinity was remarkably lower, while at *n* ≥ 6–7 all −log*K*_d_ dependencies upon *n* reached plateaus (Figure 16D). Double-stranded d(pN)_n_ showed significantly lower affinity in comparison with ss d(pN)_n_ for both sites of LA (Figure 16D). The ΔG° values characterizing the specific contribution of d(pN)_1–6_ every nucleotide unit to the total affinity for the DNA-binding site of LA were estimated (Figure 19). The spatial model of the LA–DNA complex was calculated. The protein sequence of LA has homology with five histones (H1, H2A, H2B, H3, and H4) involved in the chromatin in the interactions between themselves and their complex with DNA. It is proposed that such homology may be the main reason for the interaction of LA with DNA of chromatin, leading to a violation in its structure, as well as the proper binding of histones between themselves and with DNA [126].

## 15. Human Antibodies against DNA

The generation of autoantibodies (auto-Abs) to self-antigens such as DNA and RNA usually proceeds not only in patients with autoimmune diseases but also in healthy donors [132,133,134,135,136]. During the last three decades, it has become clear that autoantibodies from the sera of patients with various autoimmune diseases can possess DNase and RNase activities [132]. Similar to artificial abzymes produced against chemically stable analogs of transition states of catalytic reactions, natural abzymes of human blood may be auto-Abs raised directly against enzyme substrates, which act similar to haptens and mimic transient states of catalytic reactions [132,133,134,135,136]. Moreover, the second anti-idiotypic auto-Abs against catalytic centers of enzymes can also possess catalytic activities [132,133,134,135,136]. There were no data in the literature on how anti-DNA autoantibodies can recognize DNA. 

To analyze the molecular mechanism of DNA recognition, we used anti-DNA IgGs from sera of patients with multiple sclerosis [137]. The data for all used ss and ds d(pN)_n_ are summarized in Appendix A. It shows that IgGs have two DNA-binding sites; the first site corresponds to IgG heavy, while the second corresponds to the light chain of auto-Abs.

The minimal ligand of the first (1.0 × 10^−3^ M) and the second (3.3 × 10^−3^ M) IgG sites is *ortho*phosphate (Figure 20) [137]. The affinity of the IgG first binding site for deoxyribose *(K_d_* ≈ 0.14 M) and deoxyribosephosphate (d(pR)), *K_d_* ≈ 1.4 × 10^−4^ M), and the second site for deoxyribose (*K_d_* ≈ 0.13 M) and deoxyribosephosphate (4.3 × 10^−4^ M) were estimated. The transition from deoxyribosephosphate to dCMP, dTMP, and dAMP in the case of the first binding site results in the increase in the affinity respectively by a factor of ~45.2-, 73.7-, and 222.2-fold; the *K_d_* values 2.21 × 10^−2^ M, 1.36 × 10^−2^ M, and 4.5 × 10^−3^ M characterize the affinity of this site for C, T, and A bases of dNMPs. The affinity of the second site corresponding to the IgGs light chains for dCMP (2.8 × 10^−4^ M), dTMP (1.2 × 10^−5^ M), and dAMP (1.7 × 10^−4^ M) are respectively, 73.7-, 15.8-, and 269.8-fold lower than that in the case of the first DNA binding site corresponding to the heavy chains of IgGs (Appendix A). The affinity A, C, and T bases of dNMPs to the second binding site of IgGs was calculated as 0.4 M, 0.65 M, and 0.028 M. The affinity of A and C bases to the second site of IgGs is respectively 88.9- and 29.4-fold lower than that to the first site, while for T base only 2.0-fold. The affinity of the mononucleotides to the first IgG site is increased with an enhancement in the relative C < T < A bases hydrophobicity: their phosphate groups make the major contributions (1.0 × 10^−3^ M) in comparison with deoxyribose (0.14 M), but contributions of the bases to some extent comparable (from 1.36 × 10^−2^ to 4.5 × 10^−3^ M). However, the efficiencies of the bases (C < A < T) interactions with the second IgG site do not correlate with their relative hydrophobicity. Overall, both DNA binding sites of antibodies can recognize free dNMPs through the interactions with all structural elements (phosphate, sugar, and base) with the maximal contributions to the total affinity of their phosphate groups. The affinity of different homo-d(pN)_n_ to the first and the second IgG sites significantly depends on the length only for ODNs containing from one to four nucleotides (Figure 20A,B).

In addition, there was no observed correlation of the affinity with the relative hydrophobicity of ODNs bases at *n* ≥ 6–8; all d(pN)_n_, demonstrating comparable affinity (Appendix A).

The affinity of two IgG binding sites to several hetero-ODNs was also analyzed (Appendix A). The interaction of different hetero- d(pN)_n_ with the first site of IgGs is, to some extent, similar to that for homo- ODNs (Appendix A), and it is to some extent comparable with the interaction of this site with d(pA)_n_. However, several hetero-d(pN)_n_ of different sequences but of the same length show a remarkable difference in the affinity to the second DNA-binding site (for example, two 4-mer and two 9-mer ODNs; Appendix A). On average, the affinity of hetero-ODNs of different lengths to the second DNA-binding site (Appendix A) is significantly lower; for example, 13-mer hetero-ODN demonstrates 116-fold lover affinity to the second in comparison with the first site. 

We have estimated the *K*_d_ values for ds ODNs (Figure 20A,B); the minimal ligands containing two complementary chains and demonstrating remarkably higher affinity (duplex properties) toward the first site of IgGs is d(pT)_12_ × d(pA)_12_, for which the value of the melting point is higher than the used temperature (25 °C). Thus, no remarkable stabilization of short duplexes occurs by their interaction with the first site of IgGs. Ds and ss ODNs show the same affinity for the second site of IgGs. Taken together, the relative contribution of the d(pN)_n_ due to interaction with the first DNA chain is high (10^−8^–10^−9^ M). The addition of the second complementary chain leads to an average increase in the affinity only by factors of 2.4–2.6 in the case of the first and 1.0–1.3 for the second site. Thus, the contributions of the second strands are significantly less than that of the first one. This situation is very similar for all other analyzed enzymes, for which the difference between ss and ds ODNs varies two to five times (see above). Long supercoiled DNA interacts with both light and heavy chains with an affinity ~10-fold higher than that for short ODNs. The thermodynamic models were used to describe the interactions of IgGs light and heavy chains with DNA (Figure 20C,D).

## 16. Conclusions

It is known that structural elements of some low-molecular-weight ligands and substrates form very strong bonds with enzymes, 10^−5^–10^−6^ M [11,12,13,14,15,16,17,18,19,20,21]. However, a very high affinity of enzymes and proteins for specific sequences or nucleotide units of extended DNAs can be dangerous for living organisms. For example, repair enzymes show a high affinity for extended DNA, 10^−7^–10^−9^ M [11,12,13,14,15,16,17,18,19,20,21,25,26,27,54,55,56,57,58,59,60,61,69,86,87,88]. If such a high affinity is provided by the interaction of enzymes with only specific modified nucleotide units, after their removal from DNA, enzymes could be significantly inhibited by free mononucleotides. It is known that the rate of action of many enzymes is very high. Many studies of enzymes have shown that to ensure a high reaction rate, enzymes slide along DNA when searching for specific sequences and/or structural elements (specific sequences, single-stranded DNA fragments, modified nucleotides, breaks, etc.) [11,12,13,14,15,16,17,18,19,20,21,25,26,27,54,55,56,57,58,59,60,61,69,86,87,88]. Such sliding can only be achieved if the enzymes can bind to the DNA of any sequence with sufficiently high but with a close affinity for adjacent DNA fragments (isothermodynamic situation). Therefore, significant differences in the enzymes’ affinity for unspecific and specific DNA can lead to a strong suppression in the sliding speed and, as a consequence, to the decrease in reaction rate.

To prove that the most important factors are providing specificity, we first analyzed many enzymes—DNA repair, DNA replication, topoisomerization, integration, recombination, and other ones using different physiochemical approaches, including the SILC method [11,12,13,14,15,16,17,18,19,20,21,25,26,27,54,55,56,57,58,59,60,61,69,86,87,88]. It was demonstrated that a high affinity of long DNA to all enzymes and proteins is provided by forming many weak additive hydrophobic and/or van der Waals interactions with all nucleotide links covered by enzyme and protein globules. Depending on the enzyme or protein, the sum of weak unspecific additive contacts provides 5–8 orders of the enzyme affinity for specific and unspecific DNAs [11,12,13,14,15,16,17,18,19,20,21,25,26,27,54,55,56,57,58,59,60,61,69,86,87,88]. In contrast to enzymes interacting with small ligands, all interactions of enzymes with specific structural elements of long DNAs are usually weak. Their efficiencies are comparable with weak additive non-specific contacts. All specific interactions between DNA and enzymes usually provide approximately only one [11,12,13,14,15,16] and rarely about two orders of the affinity [59,92,93]. In addition, all the above-considered enzymes and proteins form complexes with DNAs and RNAs with an affinity of only about 1–1.5 orders of magnitude lower [25,26,27,53,54,55,56,57,58,59,69,86,87,88,92,93,94,95,96,97,98,101,124,125,137]. However, these DNA-dependent enzymes do not catalyze the reactions in the case of sc and ds RNA, corresponding to sequences of DNA substrates.

According to X-ray data, after DNA binding with different enzymes, there is a stage of very specific DNA and enzyme conformation adjustment (for review see [11,12,13,14,15,16]). Depending on the enzyme, there may be deformation of the DNA backbone, stretching or compression, partial or complete DNA melting, bending or kinking, eversion of nucleotides from the DNA helix, etc. [11,12,13,14,15,16]. These changes in DNA are very specific for each individual enzyme and protein. Enzyme-dependent specific changes in DNAs’ conformation are required for effective adjustment of DNA-dependent enzymes and DNAs’ reacting orbitals with an accuracy of about 10–15 degrees [22], which is possible only for specific DNAs but not RNAs [11,12,13,14,15,16]. The transition from unspecific to specific DNAs usually leads to the rise in the reaction rate (*k*_cat_) by 5–8 orders of magnitude. Taken together, the stages of enzyme-dependent adjustment of DNA conformation and direct processing of the catalysis provide the high specificity of enzymes’ actions [11,12,13,14,15,16]. Moreover, we have shown that DNA recognition by canonic enzymes and proteins with low activity and without enzymatic activities occur under the same regularities.

## Figures and Tables

**Figure 1 ijms-22-01369-f001:**
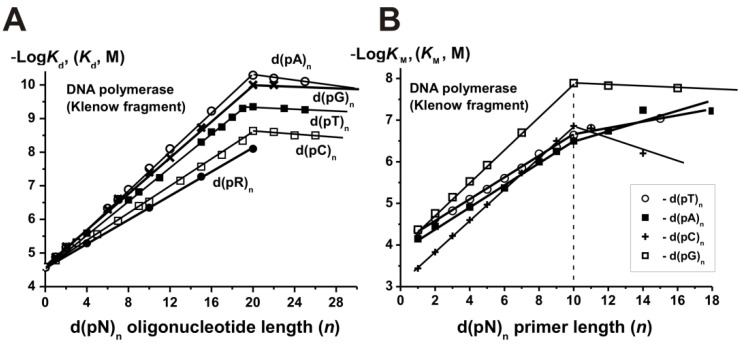
Dependencies of −Log*K*_d_ characterizing the affinity of ss d(pN)_n_ deoxyribooligonucleotides to template-binding site of DNA polymerase (Klenow fragment, *Escherichia coli*) versus their length (*n*) (**A**). All d(pN)_n_ are shown in panel A; d(pR)_n_ oligomers do not contain any bases [19,20,21]. Dependencies of −Log*K*_m_ values corresponding to different d(pN)_n_ primers on their length (*n*) in the reaction catalyzing by Klenow fragment of *E. coli.* (**B**). The *K*_m_ values in each case were determined using the poly(N) templates complementary to the primers used [23]. Figure 1A shows that the minimal ligands of the DNA polymerase template binding site are *ortho*phosphate and various dNMPs. With the lengthening of the ss d(pN)_n_, the affinity gradually increases up to 20 nucleotide units, which are covered by the enzyme globule. In addition, the growth in the affinity of ss d(pN)_n_ for the enzyme increases with an increase in the hydrophobicity of their bases: C < T < G < A. From the slopes of the curves for oligonucleotides, one can calculate the coefficient reflecting the increase in affinity with lengthening of each of the d(pN)_n_ by one nucleotide unit.

**Figure 2 ijms-22-01369-f002:**
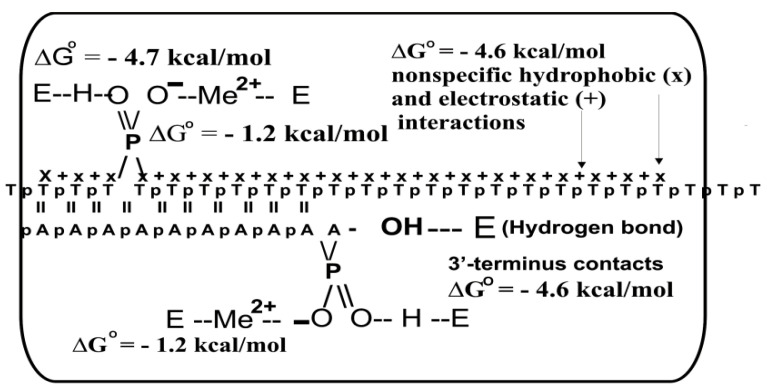
A thermodynamic model for the interaction of human DNA polymerase α with d(pT)_20_ × d(pA)_10_ [20,21]. It can be seen that the globule of DNA polymerase covers 20 nucleotide units of the template and 10 nucleotides of complementary primer. Bases of the template and 19 out of 20 internucleoside groups form weak additive contacts with the template-binding site of the enzyme (total ΔG° = −4.6 kcal/mol), while one internucleoside phosphate group of the template has strong contacts—one hydrogen bond and one electrostatic contact (total ΔG° = −4.7 kcal/mol). Only the 3′- terminal unit of the primer makes contacts with the enzyme, forming ONE hydrogen bond WITH THE 3′-OH primer group and two contacts WITH THE FIRST from the 3′-end internucleoside phosphate group (total ΔG° = −4.6 kcal/mol). The rest of the primer nucleotide units interact only with the complementary template through the formation of Watson–Crick hydrogen bonds.

**Figure 3 ijms-22-01369-f003:**
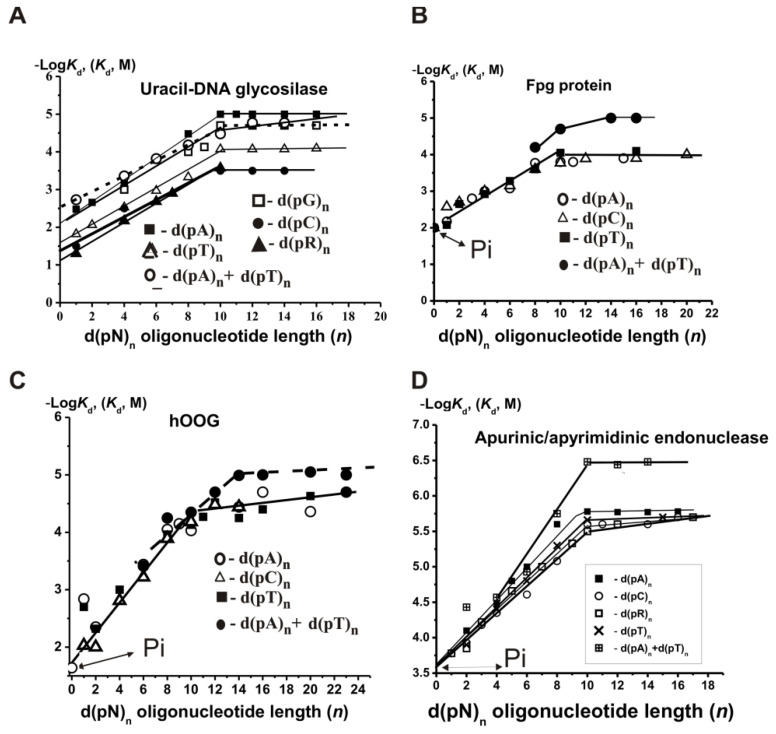
Dependencies of –Log*K*_d_ for ss and ds d(pN)_n_ deoxyribooligonucleotides versus their length (*n*) in the case of uracil DNA glycosylase (**A**) [26,27], Fpg protein (**B**) [54,55,56,57], hOGG1 (**C**) [59], and apurinic/apyrimidinic endonuclease (**D**) [69]. All oligonucleotides used are shown in panel (**A**–**D**); d(pR)_n_ oligomers do not contain any bases.

**Figure 4 ijms-22-01369-f004:**
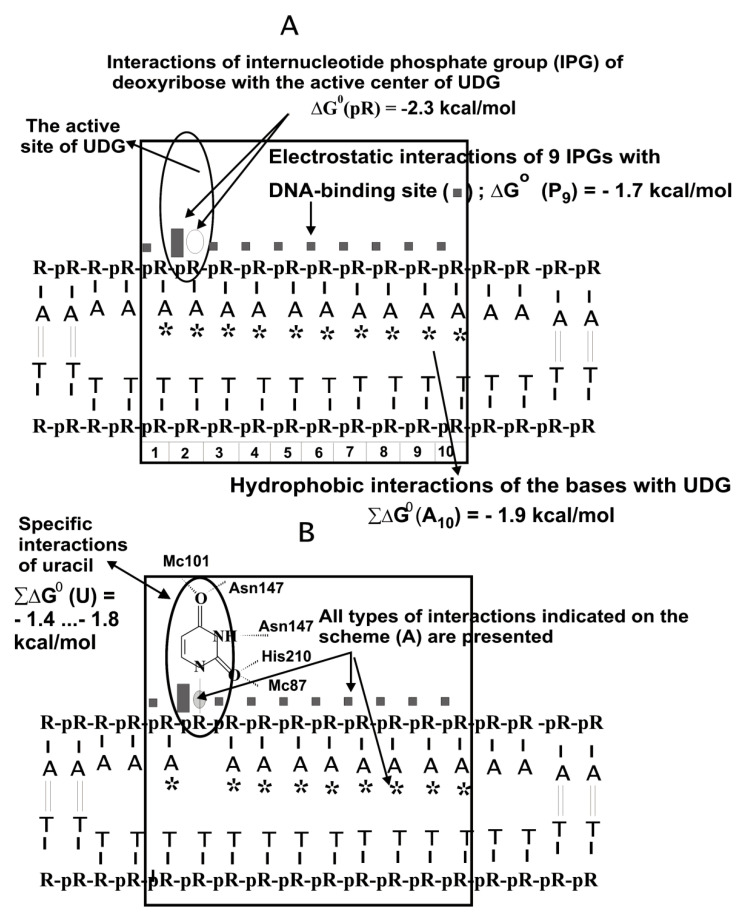
The thermodynamic model of human UDG interaction with unspecific (**A**) and specific DNA containing dU nucleotide unit (**B**). The ΔG° value characterizing of five specific hydrogen bonds between UDG and U-base was evaluated to be approximately –1.3 kcal/mol (**B**), while unspecific electrostatic (■) and hydrophobic (★) interactions provide ΔG° = −5.9 kcal/mol (**A**) [27].

**Figure 5 ijms-22-01369-f005:**
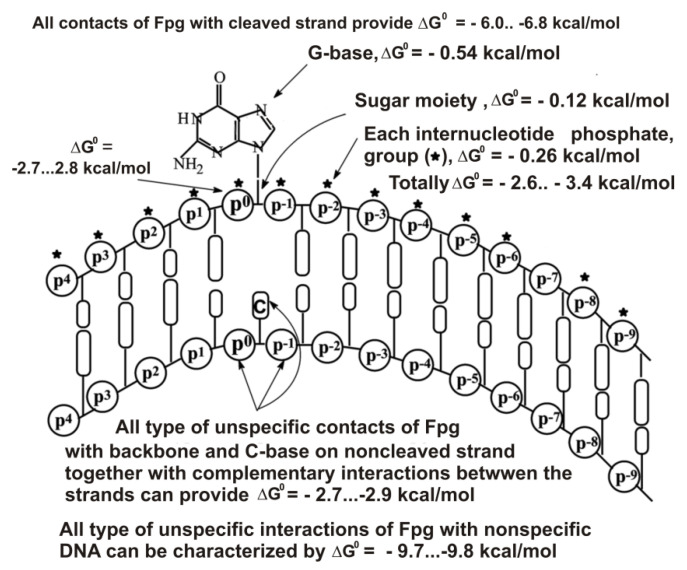
The thermodynamic model of *E. coli* Fpg interaction with unspecific DNA. All types of unspecific contacts are shown on the Panel [57]. The enzyme forms weak additive contacts only with internucleoside phosphate groups of DNA and there are contacts only of the active center with one of the DNA bases.

**Figure 6 ijms-22-01369-f006:**
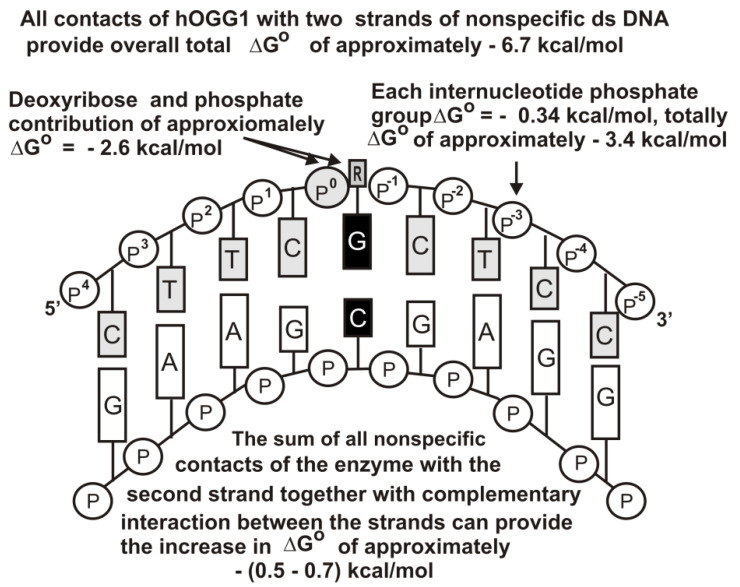
The thermodynamic model of hOGG1 interaction with unspecific DNA. All types of unspecific contacts are shown on the Panel [59]. The enzyme forms weak additive contacts only with internucleoside phosphate groups of DNAs except for one deoxyribose phosphate.

**Figure 7 ijms-22-01369-f007:**
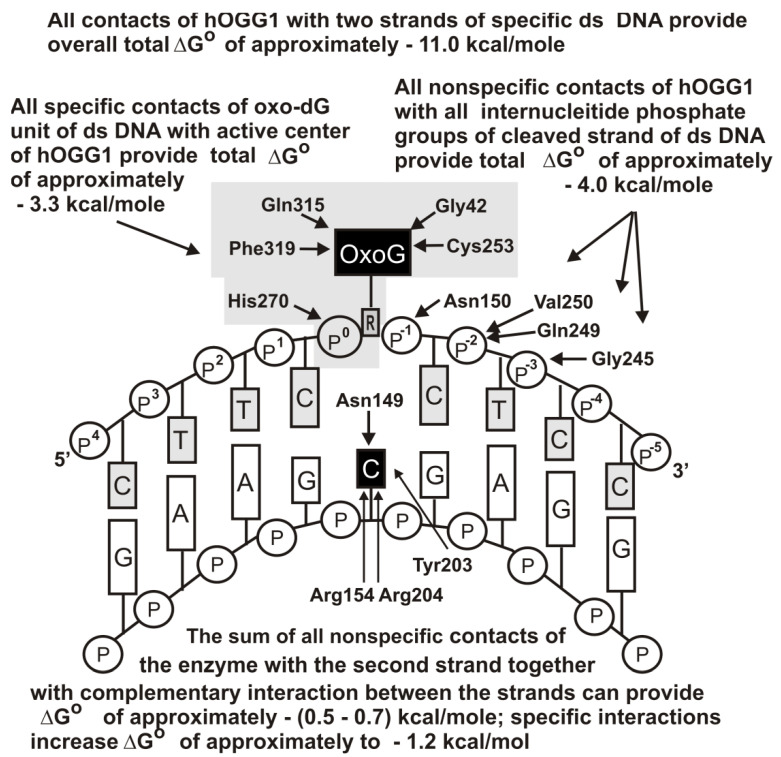
The thermodynamic model of hOGG1interaction of with specific DNA containing 8-oxoG. All types of unspecific contacts are shown on the Panel [59].

**Figure 8 ijms-22-01369-f008:**
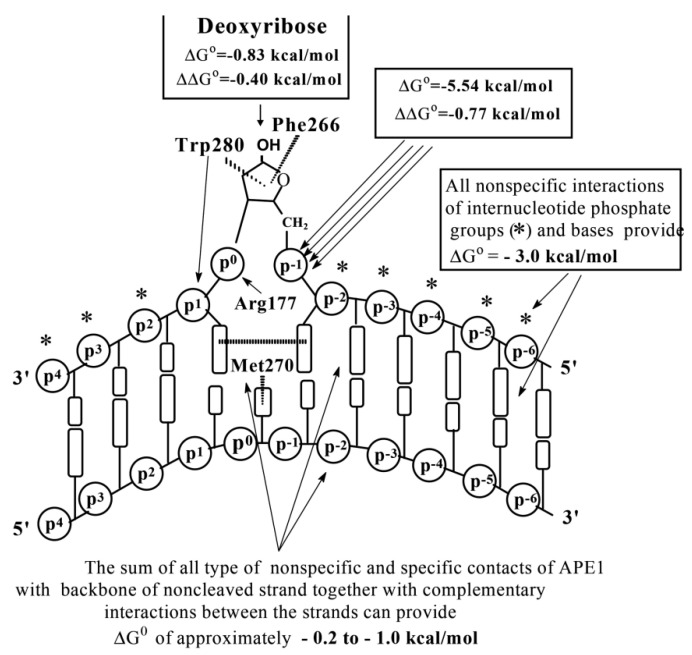
The thermodynamic model of apurinic/apyrimidinic endonuclease (APE1) interaction with specific DNA containing abasic (AP)-site. All types of contacts are shown on the Panel [69].

**Figure 9 ijms-22-01369-f009:**
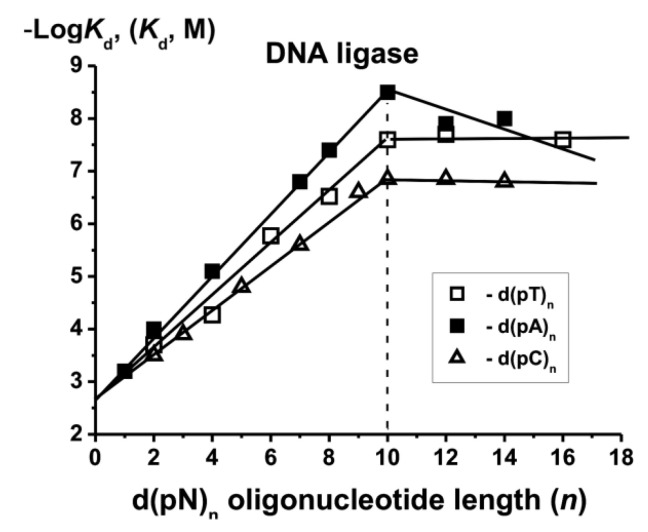
Dependencies of −Log*K*_d_ for ss and ds d(pN)_n_ deoxyribooligonucleotides versus their length (*n*) in the case of DNA ligase [12,13].

**Figure 10 ijms-22-01369-f010:**
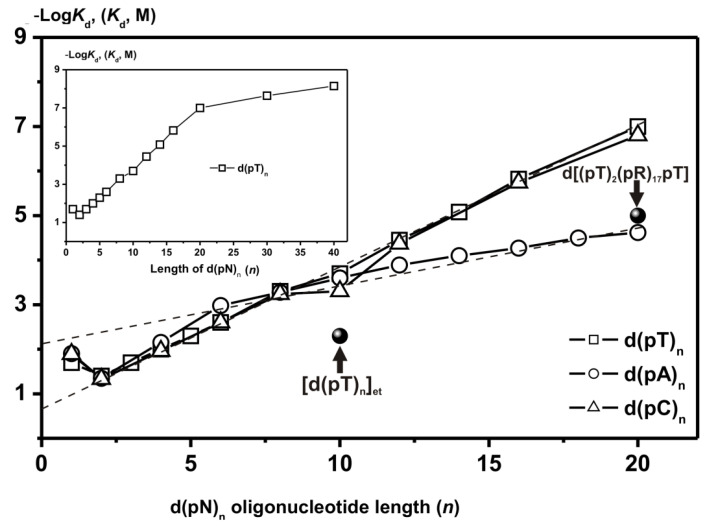
Dependencies of −Log*K*_d_ for ss and ds d(pN)_n_ deoxyribooligonucleotides versus their length (*n*) in the case of the second site of RecA protein [86,87,88]. For comparison, the inset shows the Log-dependences of the *K*_d_ values characterizing the affinity of the first site of the RecA filament to d(pT)_n_. All types of ODNs including those containing not many bases [d(pT)_2_(pR)_17_pT]) and completely ethylated at internucleoside phosphate groups ([d(pT)_n_]_et_ are shown on the Panel.

**Figure 11 ijms-22-01369-f011:**
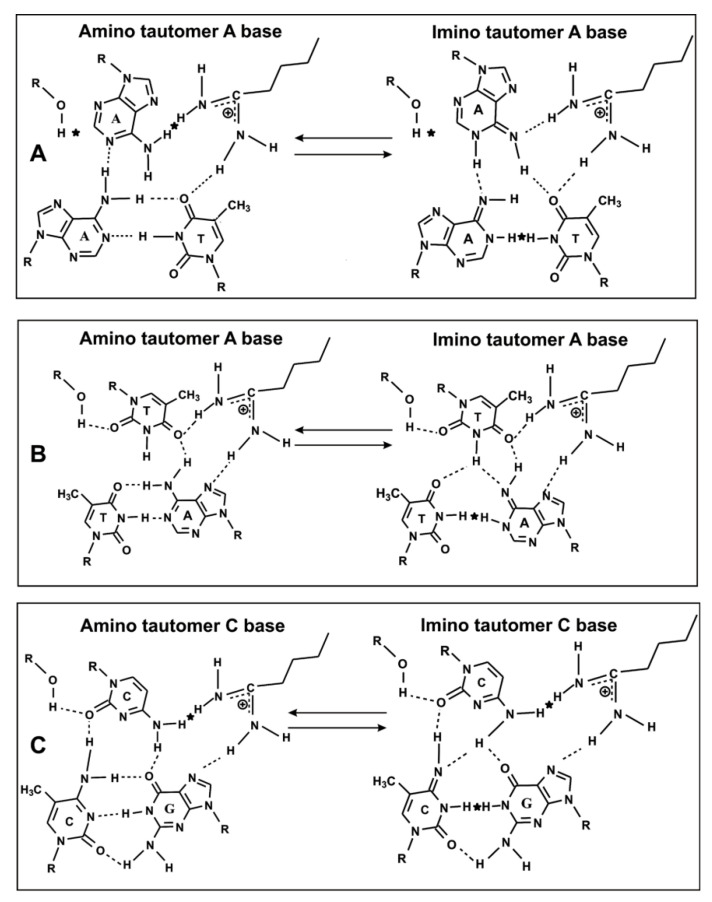
The hypothetical mechanism of homologous exchange catalyzed by RecA, proposed in [87,88]. The binding of ss DNA and the formation of a filament is accompanied by the formation of contacts of the Tyr and Arg RecA residues of the protein with one of the bases of DNA (**A**–**C**). Subsequent binding of the duplex by the RecA-ss DNA complex and the search for homology is provided due to the formation of specific hydrogen bonds during the formation of the R-DNA triplex. In this case, the RecA protein can catalyze the destruction of Watson–Crick bonds in the duplex due to a shift in the amino–imine equilibrium after the formation of contacts from the one of base with the amino acid residue of the DNA recognition center RecA. The destruction of hydrogen bonds in the original duplex leads to the formation of new ds DNA. The impossibility of hydrogen bonding is indicated using (★). Tyr and Arg are used in the model as the most appropriate amino acid residues for this role.

**Figure 12 ijms-22-01369-f012:**
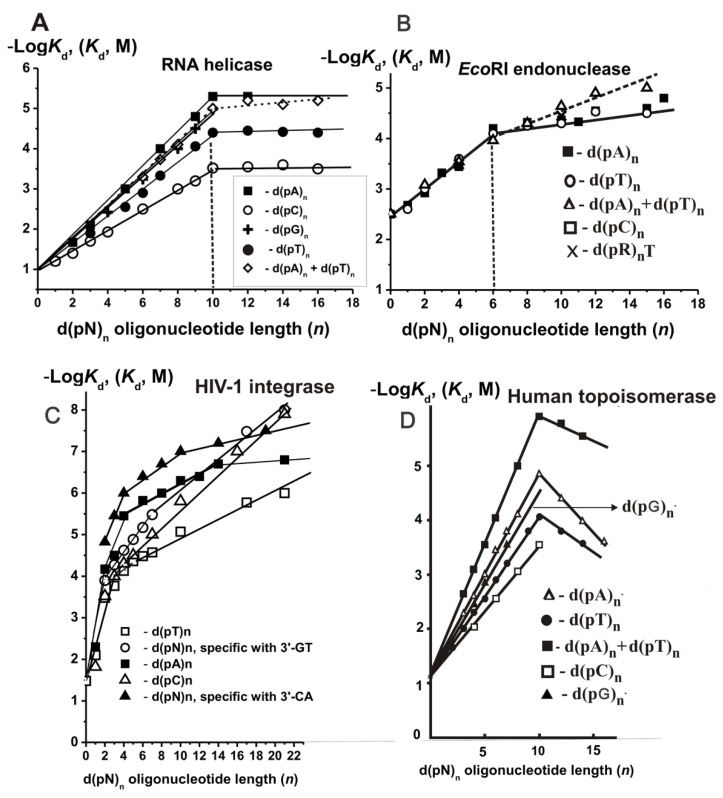
Dependencies of –Log*K*_d_ for ss and ds d(pN)_n_ deoxyribooligonucleotides versus their length (*n*) in the case of RNA–DNA helicase (**A**) [12,13], *Eco*R1 endonuclease (**B**) [92,93], HIV-1 integrase (**C**) [94,95], and human topoisomerase I (**D**) [96,97,98]. All types of enzymes and ODNs are shown on the panels; d(pR)_n_pT oligomers contain only one T base.

**Figure 13 ijms-22-01369-f013:**
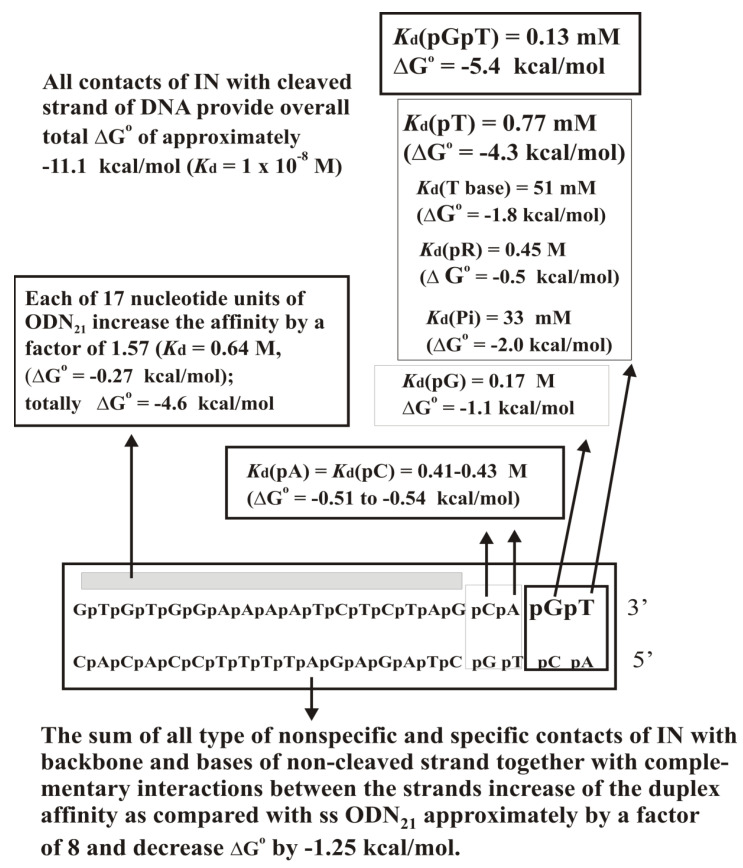
Thermodynamic model of HIV-1 IN interaction of with cognate ds GT-ODN_21_ containing 3’-terminal-specific GT dinucleotide. ΔG° values characterizing different contacts between the enzyme and DNA strands are shown [95].

**Figure 14 ijms-22-01369-f014:**
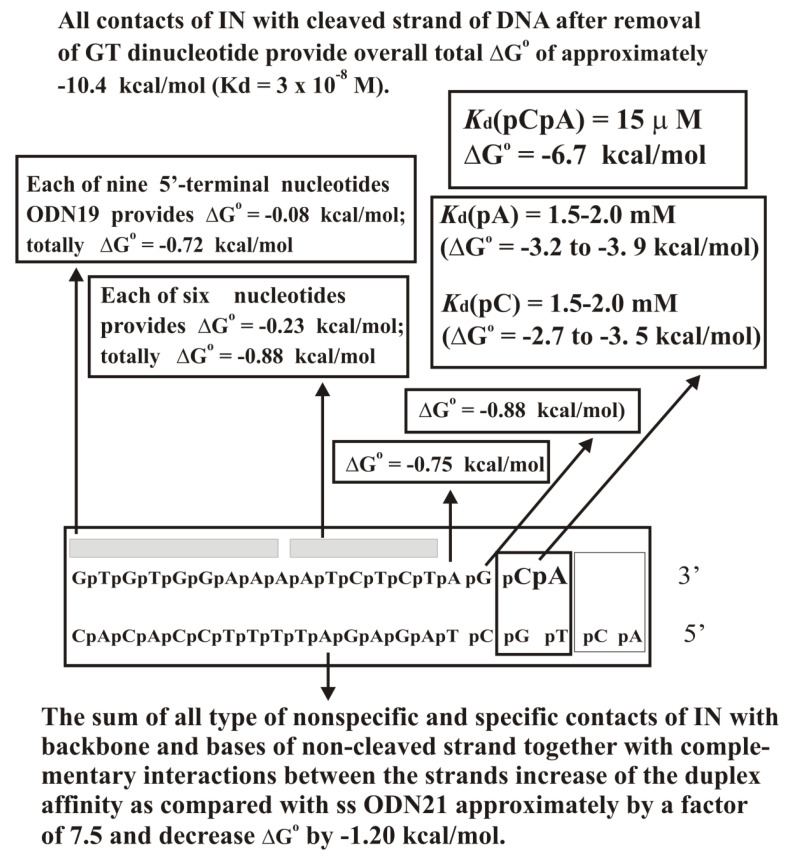
Thermodynamic model of HIV-1 IN interaction of with cognate ds CA-specific ODN_19_ after removal of 3’-terminal-specific GT dinucleotide from the cleavable chain of ds ODN_21_ [95]. ΔG° values characterizing various contacts between the enzyme and DNA strands are shown.

**Figure 15 ijms-22-01369-f015:**
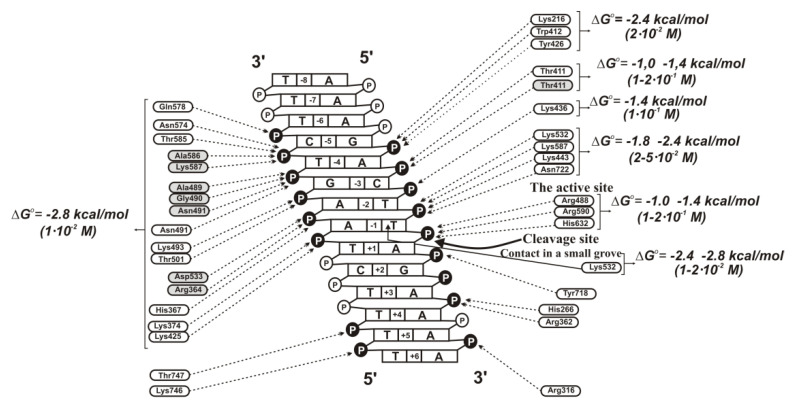
Schematic representation of the relative efficiency of different contacts formed by amino acid residues of human Topo I with internucleoside phosphate groups and the T-base of a specific sequence, revealed by X-ray analysis. The data on the right and left show approximate *K*_d_ values and ∆G° estimated using the SILC method [11,12,13].

**Figure 16 ijms-22-01369-f016:**
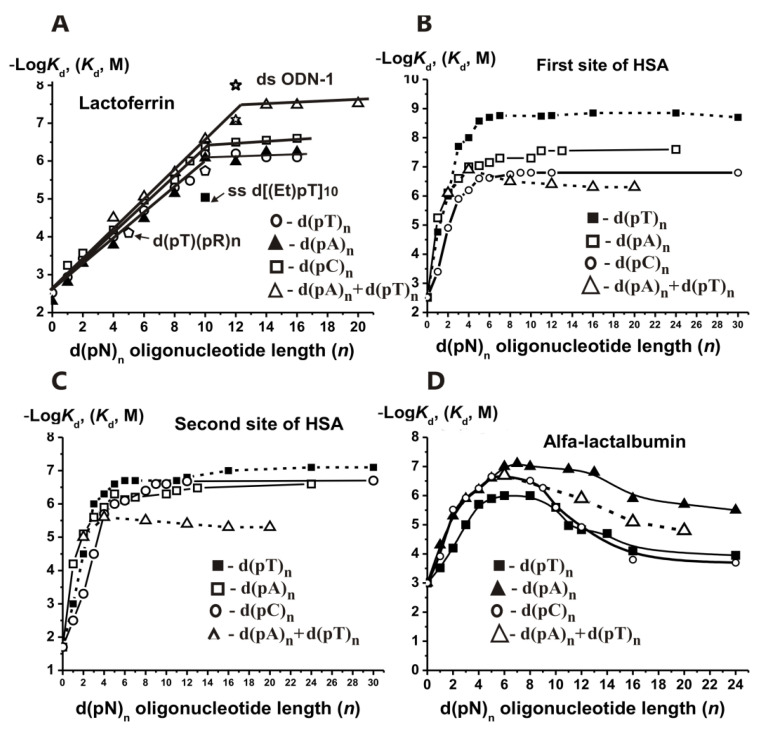
Dependencies of −Log*K*_d_ for ss and ds d(pN)_n_ deoxyribooligonucleotides versus their length (*n*) in the case of human lactoferrin (**A**) [124], first (**B**), and second (**C**) DNA-binding sites of human albumin [125], and human α-lactalbumin (**D**) [126]. All types of enzymes and ODNs are shown on the Panels; d(pT)(pR)_n_ oligomers contain only one T base, d[(Et)pT]_10_ is oligonucleotide with ethylated internucleoside phosphate groups.

**Figure 17 ijms-22-01369-f017:**
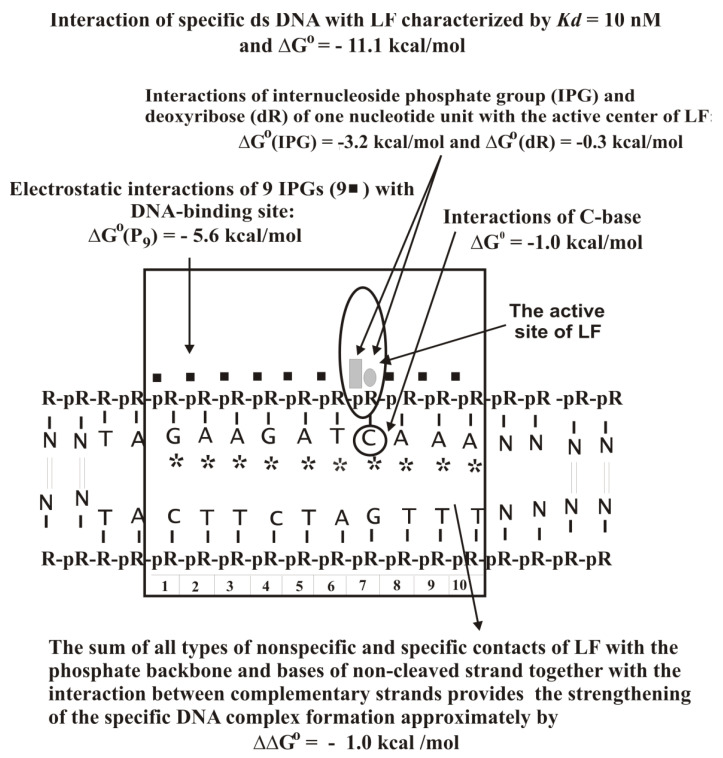
Thermodynamic model of human lactoferrin interaction with specific oligonucleotide. ΔG° values characterizing various contacts between the protein and DNA are shown [124].

**Figure 18 ijms-22-01369-f018:**
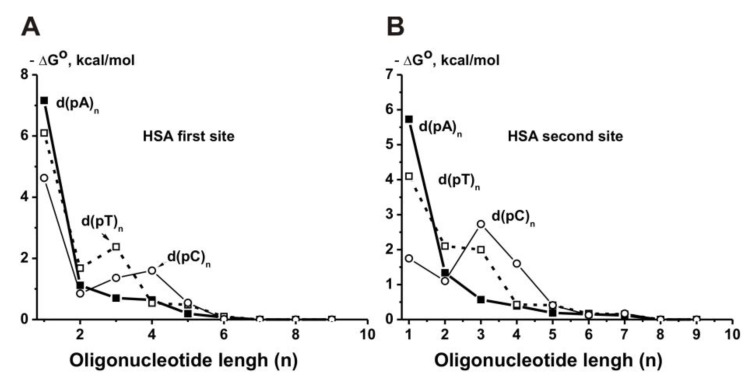
Thermodynamic values (−ΔG°) characterizing a relative contribution of various mononucleotides of ss ODNs to the total affinity of these d(pN)_8_ for the first (**A**) and the second (**B**) sites of human serum albumin (has). Different ss d(pN)_n_ are shown on panels (**A**,**B**) [125].

**Figure 19 ijms-22-01369-f019:**
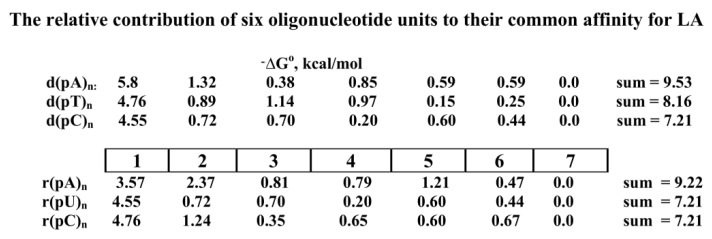
Thermodynamic model (−ΔG° values) characterizing a relative contribution of different mononucleotides of ss d(pN)_7_ and ribo(pN)_7_ to the total affinity of these (pN)_7_ for human milk α-lactalbumin (LA) [126]. Different ss (pN)_7_ are shown on the Panel.

**Figure 20 ijms-22-01369-f020:**
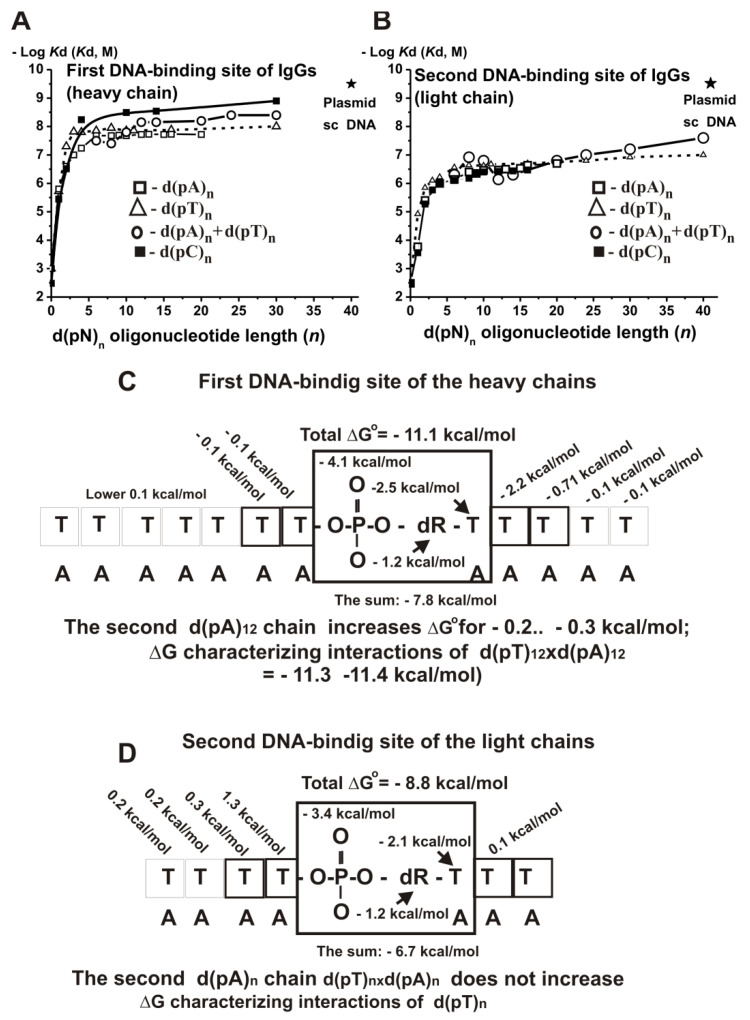
The −Log dependences of *K*_d_ values for d(pN)_n_ corresponding to the first (**A**) and second (**B**) sites of anti-DNA IgGs upon ODNs length [137]. Different ss and ds ODNs are shown on the Panels (**A**,**B**). The average error in the complex formation from two independent experiments did not exceed 7–10%. Thermodynamic models of d(pT)_n_ interaction with the heavy (**C**) and light (**D**) chains of IgGs. Values ∆G° are given on the Panels (**C**,**D**).

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
