# Peer review of "How Enzymes, Proteins, and Antibodies Recognize Extended DNAs; General Regularities"

_ijms, 2021, doi:10.3390/ijms22031369_

Round 1

Reviewer 1 Report

This is a comprehensive review covering work from GA Nevinsky from the mid 1990's. The review states (lines 47-49), “Therefore, we have developed a new approach [stepwise increase in ligand complexity (SILC)] to estimate the relative contributions of all individual nucleotides or specific sequences as well as their various structural elements to an enzyme’s affinity for long DNAs [11-16].” This basic approach (not new) has been seemingly employed by the author since 1987, and the author has published many reviews over the years on his work. That brings into question the need, goal, and focus of the extant review.

I question if the review is too long and too hard to follow. It is comprehensive and well organized - basically chronologically. Each section usually begins with a well-written excellent description of the enzyme or binding protein. Then the author delves into many specific details that may be unnecessary for a review. I found reading the specifics and details laborious. One also questions the need for 26 supplemental tables from previous publications. Perhaps it is useful to compile them in one place? Sections then end with a summary of the results. One suggestion for shortening and focusing the review would be to omit so many details as they are in supplemental tables and referenced papers, and simply focus on salient conclusions.

It appears most figures are previously published. Figures of -LogKd vs. n are easy to follow but I find the thermodynamic model figures difficult to comprehend and understand. While the author has been publishing these types of figures, which do contain a great deal of information, for quite some time, I feel that persons reviewing this body of work for the first time will struggle with them. I would suggest that the author spend a good deal of time walking a reader through one of these early (Figure 2 or 4). Note that the legends do not explain the figures, nor are they explained in the text. Description of what to focus on or how to look at these thermodynamic models may help a reader that has not seen these previously. This could be done in the Figure legends.

Some overall summary figure that helps interpret conclusions might be useful.

As the author has written numerous reviews over this material, the present review might focus more on the latter proteins that can bind nucleic acids and compare this with the enzymes reviewed previously, comparing and contrasting modes of interaction.

There are a number of sentences where the English warrants attention. In addition there are quite a few instances of errors, especially in reference numeration. Reference numbers in the review should be checked carefully.
A few examples:
Line 387: (83, 83) should read (82, 83)
Line 414: Table 87?
There are also errors in formatting: perhaps equation (3), cleavage sites, especially lines 582-588, that I’m sure would be cleaned up in a final draft.

What is meant by pseudo Watson-Crick bonds? Line 194-5

Author Response

All fixes in the article are marked in red.

  1. This is a comprehensive review covering work from GA Nevinsky from the mid 1990's. The review states (lines 47-49), “Therefore, we have developed a new approach [stepwise increase in ligand complexity (SILC)] to estimate the relative contributions of all individual nucleotides or specific sequences as well as their various structural elements to an enzyme’s affinity for long DNAs [11-16].” This basic approach (not new) has been seemingly employed by the author since 1987, and the author has published many reviews over the years on his work. That brings into question the need, goal, and focus of the extant review.

I question if the review is too long and too hard to follow. It is comprehensive and well organized - basically chronologically. Each section usually begins with a well-written excellent description of the enzyme or binding protein. Then the author delves into many specific details that may be unnecessary for a review. I found reading the specifics and details laborious. One also questions the need for 26 supplemental tables from previous publications. Perhaps it is useful to compile them in one place? Sections then end with a summary of the results. One suggestion for shortening and focusing the review would be to omit so many details as they are in supplemental tables and referenced papers, and simply focus on salient conclusions.

Answer: Sorry, but unfortunately, I cannot agree with you. It should be noted that this review, in a special sense, has a philosophical character. Unfortunately, the understanding of how enzymes and proteins recognize extended nucleic acid molecules is currently very far from correct. In all previous reviews, some shortened data concerning  one or another part of the review was described and there was no summary generalization. It seemed unlikely that anyone would read many of our articles and reviews and would be able to combine everything into a single whole - general patterns of recognition of extended nucleic acids. The purpose of this review was in an accessible way in one review, using a large number of examples, to explain that recognition is based on general patterns, namely, that the high affinity of any extended DNA molecules for proteins, enzymes, antibodies is provided primarily by a multitude of weak additive interactions with all structural elements of individual nucleotide links. At the same time, the contribution of specific interactions is relatively small. On the one hand, this is so, these are general patterns. However, these general patterns are superimposed on the specific features of each enzyme or protein. According to X-ray data, after DNA binding with different enzymes, there is a stage of very specific DNA and enzyme conformation adjustment. Depending on the enzyme, there may be very different deformations of DNA backbone, stretching or compression, partial or complete DNA melting, bending or kinking, eversion of nucleotides from the DNA helix, etc. Such changes in DNA are very specific and very important for each individual enzyme and protein. These data on specific changes in DNA conformation, which are also important for understanding the general patterns of recognition, are also reviewed in the review. Taking this into account, in order to understand all possible individual aspects of specific DNA recognition by each individual enzyme, it is necessary to demonstrate the details-features of recognition by each protein and only then combine all the data into a single picture of different recognition options. This turned out to be possible only with a relatively detailed analysis of each protein and enzyme, which allows the reader, based on the entire set of data and specific details for each enzyme, to see the way to study the peculiarities of DNA recognition by different enzymes and proteins. In addition, a general understanding of the recognition patterns can only be shown by giving a complete picture of DNA recognition using the example of sequence-specific and non-specific proteins and enzymes.

Another purpose of the review: it was very important to explain with the help of what adequate methods of analysis it is possible to identify the basic patterns of DNA recognition. Unfortunately, the understanding of physicochemical approaches to the study of recognition mechanisms is mainly narrowed and weak. Taking this into account, an attempt was made to explain the essence and possibilities of the SILC method of to show that it does not matter which enzyme, protein you are studying - the general patterns are the same. I did not include our data of protein recognition by enzymes and antibodies in the review, since the review is devoted to DNA recognition. However, recognition of proteins by enzymes and antibodies also occurs in accordance with the same patterns as DNA. From my point of view, only a detailed explanation of the features of the method can help researchers to understand how it can be applied.

  1. It appears most figures are previously published. Figures of -LogKd vs. n are easy to follow but I find the thermodynamic model figures difficult to comprehend and understand. While the author has been publishing these types of figures, which do contain a great deal of information, for quite some time, I feel that persons reviewing this body of work for the first time will struggle with them. I would suggest that the author spend a good deal of time walking a reader through one of these early (Figure 2 or 4). Note that the legends do not explain the figures, nor are they explained in the text. Description of what to focus on or how to look at these thermodynamic models may help a reader that has not seen these previously. This could be done in the Figure legends. Some overall summary figure that helps interpret conclusions might be useful.

Answer:

Sorry, but the contribution of each of the links of extended DNA to affinity can be clearly shown only with the help of thermodynamic models. Figures of -LogKd vs. n  could not show mutual picture of recognition. For a simple understanding of these pictures, the introduction gives the following information.

The Gibbs free energy (∆Go) for the complexes of ligands with enzymes and proteins are usually equal to the sum of the ∆Go values ​​for all individual contacts formed: ∆Gosum (corresponding to all n contacts) = ∆Go1 + ∆Go2 +… + ∆Gon (1), where all ∆Go1-n correspond to all individual contacts of the ligand; ∆Go for each individual contact = - RT × lnKd [22]. The total Kd value characterizing the formation of any protein-enzyme complex with the ligand is the product of Kd values ​​for all individual contacts: Kd = Kd(1) × Kd(2) × …… Kd(n) (2). From the Gibbs free energy values presented in the models, it is immediately clear what contribution to the affinity is made by each of the nonspecific or specific DNA nucleotide units.

In legends of Figure 1 were added some explanation.

  1. As the author has written numerous reviews over this material, the present review might focus more on the latter proteins that can bind nucleic acids and compare this with the enzymes reviewed previously, comparing and contrasting modes of interaction.

Answer:

Sorry, if you focus on recent research, the picture of the features in the case of each enzyme disappears. And as a result, the possible variety of manifestations of the same and different features of various enzymes will disappear despite the fact that there are common patterns of DNA recognition for all of them. This was the idea of this review to bring together common patterns and all variants of the diversity of DNA recognition by different enzymes and proteins.

  1. There are a number of sentences where the English warrants attention. In addition there are quite a few instances of errors, especially in reference numeration. Reference numbers in the review should be checked carefully.
    A few examples:
    Line 387: (83, 83) should read (82, 83)
    Line 414: Table 87?
    There are also errors in formatting: perhaps equation (3), cleavage sites, especially lines 582-588, that I’m sure would be cleaned up in a final draft.

Answer:

It was corrected

  1. What is meant by pseudo Watson-Crick bonds? Line 194-5

Answer:

In the literature there is such a term as pseudo-Watson-Crick bonds. This term refers to the formation bonds between groups of DNA bases, which enter into Watson-Crick interactions with honey bases, contacts with a protein. However, taking into account your comment, I have removed the term.

Sorry, the purpose of the review was not only to make the final conclusions comprehensible. It was important to show what path exists to these conclusions, with the help of what methods and approaches it is possible to prove that these conclusions are correct, fair and unmistakable. In order to use the SILC analysis method, if necessary, it is important to understand the details of such analysis. So in the appendix there are Tables of values ​​of the dissociation constants of oligonucleotides of different lengths with different enzymes and proteins. If a person understands everything without these Tables, he/she may not look at the applications. However, if one examines the recognition of extended DNA by any protein or enzyme, then on the basis of these Tables one can estimate in what range of ligand concentrations one should work. In addition, given the pier. the mass of the protein can be estimated how many DNA units can cover the enzyme globule. These tables are also important for understanding that the minimal ligands of all enzymes are orthophosphate and mononucleotides and that their affinity depending on the protein can vary greatly - from 0.1 M to 10-5 M. An attempt was made to write the review so that it would be useful as for scientific advisors and graduate students and students in their practical work.

Many thanks for all your useful comments, they allowed us to make important additions and corrections to the article.

Sincerely

Prof. Georgy A. Nevinsky

Reviewer 2 Report

This review paper by Nevinsky discusses the data on SILC analysis of recognition between nucleic acid and various enzymes.
The manuscript is well-written. I believe that the manuscript can be accepted after minor comments below are addressed.

Minor comments:

For reader of this journal, please add schematic diagram or cartoon for stepwise increase in ligand complexity (SILC) as a Figure 1 in the introduction section.
Also, it should be compared to the schematic diagram or cartoon of the traditional method, because SILC is main topic in this review.

For “Kd” in all figures, use the same description: “K” for capital italic and “d” for subscript. This is very important in discussing physical chemistry matters.
The resolution of figure also should be improved.

page 2, line 87,
Supplementary Table 1 → Supplementary Table S1

page 3, line 129,
Supplementary Table 2 → Supplementary Table S2

page 4, line 165,
Supplementary Table 3 → Supplementary Table S3

page 4, line 143, end of caption for Figure 2,
If “Figure 17” is typo, it should be corrected.
If it is correct, please add some explanatory paragraph.

page 7, line 230,
Supplementary Table 4 → Supplementary Table S4

page 7, line 232,
Supplementary Table 5 → Supplementary Table S5

page 8, in Figure 5, the structure of guanine is not correct. It should be corrected.

page 11, in Figure 8, the structure of deoxyribose is not correct.
The structure should be corrected to 5’-3’, not 5’-2’.

page 15, in Figure 10A,
The “T” is behind the carbon in 5 position of the thymine nucleobase.
Please delete it.

page 15, line 414
Where is Table 87? If this is typo, it should be corrected.

page 15, line 449~451
The arrow indicating the points of cleavage is misaligned. It should be corrected.

page 17, line 485,
Supplementary Table 12 → Supplementary Table S12

page 19, in Figure 12,

page 20, line 581~588,
It should be corrected appropriately.

page 23, line 675, about “DNA-binding”
Font and font size should be corrected.

page 23, line 691,
“(D)” should be corrected to bold type.

page 24, line 703,
Supplementary Table 21 → Supplementary Table S21

page 24, in Figure 16
Please correct to subscript “d” at “by Kd = 10 nM”.

page 25, line 740,
Supplementary Table 22 → Supplementary Table S22

page 26, line 765,
Supplementary Table 24 → Supplementary Table S24

page 27, line 806,
Supplementary Table 25 → Supplementary Table S25

page 28, line 826, 828, 829, 832, 833,
Supplementary Table 24, 25, 26 → Supplementary Table S24, S25, S26

Please add Topo I and PCA to Abbreviations sections.

page 30, line 927~928,
Please delete Hyperlink of the title in Ref. 13.

Author Response

This review paper by Nevinsky discusses the data on SILC analysis of recognition between nucleic acid and various enzymes. The manuscript is well-written. I believe that the manuscript can be accepted after minor comments below are addressed.
Minor comments:

1) For reader of this journal, please add schematic diagram or cartoon for stepwise increase in ligand complexity (SILC) as a Figure 1 in the introduction section.

Answer: All fixes in the article are marked in red. Scheme inserted in the introduction text

  1. For “Kd” in all figures, use the same description: “K” for capital italic and “d” for subscript. This is very important in discussing physical chemistry matters.
    The resolution of figure also should be improved.

Answer:

It was done and all figures were improved

  1. Supplementary Table 1 → Supplementary Table S1

page 3, line 129,
Supplementary Table 2 → Supplementary Table S2

page 4, line 165,
Supplementary Table 3 → Supplementary Table S3

page 4, line 143, end of caption for Figure 2,
If “Figure 17” is typo, it should be corrected.
If it is correct, please add some explanatory paragraph.

page 7, line 230,
Supplementary Table 4 → Supplementary Table S4

page 7, line 232,
Supplementary Table 5 → Supplementary Table S5

page 8, in Figure 5, the structure of guanine is not correct. It should be corrected.

Answer: All were corrected

  1. page 11, in Figure 8, the structure of deoxyribose is not correct. groupThe structure should be corrected to 5’-3’, not 5’-2’.

Answer: Sorry, but on Fig. 8 there is no error - just the CH2 group is placed not on the left but on the right from O-atom. To make it clear in place of the missing base, I put the OH

  1. page 15, in Figure 10A,
    The “T” is behind the carbon in 5 position of the thymine nucleobase.
    Please delete it.

Answer: It was corrected

  1. page 15, line 414
    Where is Table 87? If this is typo, it should be corrected.

Answer: It was corrected

  1. page 15, line 449~451
    The arrow indicating the points of cleavage is misaligned. It should be corrected.

Answer: It was corrected

page 17, line 485,
Supplementary Table 12 → Supplementary Table S12

page 19, in Figure 12,

page 20, line 581~588,
It should be corrected appropriately.

page 23, line 675, about “DNA-binding”
Font and font size should be corrected.

page 23, line 691,
“(D)” should be corrected to bold type.

page 24, line 703,
Supplementary Table 21 → Supplementary Table S21

page 24, in Figure 16
Please correct to subscript “d” at “by Kd = 10 nM”.

page 25, line 740,
Supplementary Table 22 → Supplementary Table S22

page 26, line 765,
Supplementary Table 24 → Supplementary Table S24

page 27, line 806,
Supplementary Table 25 → Supplementary Table S25

page 28, line 826, 828, 829, 832, 833,
Supplementary Table 24, 25, 26 → Supplementary Table S24, S25, S26

Please add Topo I and PCA to Abbreviations sections.

page 30, line 927~928,
Please delete Hyperlink of the title in Ref. 13.

Answer:  All were corrected

Many thanks for all your useful comments, they allowed us to make important additions and corrections to the article.

Sincerely

Prof. Georgy A. Nevinsky

Round 2

Reviewer 1 Report

The revised version of this Review is greatly improved with the relatively minor additions.

The Author has made the case for a comprehensive review rather than shortening the manuscript.

I note a few places that may warrant editing:

Line 8: Should 'expended DNA' read "extended DNA"?

Line 20: Please clarify the meaning or inclusion of "directly catalysis."  The meaning is not clear.  Or should there be a comma after directly. 

Line 132: I believe that 'have' should be replaced by 'has' 

Lines 162-164:  

The sentence at lines 162-164 is not entirely clear -

Maybe edit to read? (EDITS IN CAPS):

Only the 3’- terminal unit of the primer makes contacts with the enzyme, forming ONE hydrogen bond WITH THE 3’-OH primer group and two contacts WITH to the first BASE from the 3’-end of the internucleoside phosphate group (total ΔG° = - 4.6 kcal/mol).

Line 184:  Delete 'of'

Author Response

The revised version of this Review is greatly improved with the relatively minor additions.

The Author has made the case for a comprehensive review rather than shortening the manuscript.

I note a few places that may warrant editing:

Line 8: Should 'expended DNA' read "extended DNA"?

Answer: it was corrected

Line 20: Please clarify the meaning or inclusion of "directly catalysis."  The meaning is not clear.  Or should there be a comma after directly. 

Answer: it was corrected

Line 132: I believe that 'have' should be replaced by 'has' 

Answer: it was corrected

Lines 162-164:  

The sentence at lines 162-164 is not entirely clear -

Maybe edit to read? (EDITS IN CAPS):

Only the 3’- terminal unit of the primer makes contacts with the enzyme, forming ONE hydrogen bond WITH THE 3’-OH primer group and two contacts WITH to the first BASE from the 3’-end of the internucleoside phosphate group (total ΔG° = - 4.6 kcal/mol).

Answer: it was corrected

Thanks a lot for the remarks

Sincerely

Georgy A. Nevinsky